# Characteristics and impact of Long Covid: Findings from an online survey

**Nida Ziauddeen**[1,2]*, **Deepti Gurdasani**[3], **Margaret E. O'Hara**[4], **Claire Hastie**[4], **Paul Roderick**[1], **Guiqing Yao**[5], **Nisreen A. Alwan**[1,2,6]*

**1** School of Primary Care, Population Sciences and Medical Education, Faculty of Medicine, University of Southampton, Southampton, United Kingdom, **2** NIHR Applied Research Collaboration Wessex, Southampton, United Kingdom, **3** William Harvey Research Institute, Queen Mary University of London, London, United Kingdom, **4** Patient contributor, Long Covid Support: www.longcovid.org, London, United Kingdom, **5** Department of Health Science, University of Leicester, Leicester, United Kingdom, **6** NIHR Southampton Biomedical Research Centre, University of Southampton and University Hospital Southampton NHS Foundation Trust, Southampton, United Kingdom

* Nida.Ziauddeen@soton.ac.uk (NZ); N.A.Alwan@soton.ac.uk (NAA)

## Abstract

### Background

Long Covid is a public health concern that needs defining, quantifying, and describing. We aimed to explore the initial and ongoing symptoms of Long Covid following SARS-CoV-2 infection and describe its impact on daily life.

### Methods

We collected self-reported data through an online survey using convenience non-probability sampling. The survey enrolled adults who reported lab-confirmed (PCR or antibody) or suspected COVID-19 who were not hospitalised in the first two weeks of illness. This analysis was restricted to those with self-reported Long Covid. Univariate comparisons between those with and without confirmed COVID-19 infection were carried out and agglomerative hierarchical clustering was used to identify specific symptom clusters, and their demographic and functional correlates.

### Results

We analysed data from 2550 participants with a median duration of illness of 7.6 months (interquartile range (IQR) 7.1–7.9). 26.5% reported lab-confirmation of infection. The mean age was 46.5 years (standard deviation 11 years) with 82.8% females and 79.9% of participants based in the UK. 89.5% described their health as good, very good or excellent before COVID-19. The most common initial symptoms that persisted were exhaustion, chest pressure/tightness, shortness of breath and headache. Cognitive dysfunction and palpitations became more prevalent later in the illness. Most participants described fluctuating (57.7%) or relapsing symptoms (17.6%). Physical activity, stress, and sleep disturbance commonly triggered symptoms. A third (32%) reported they were unable to live alone without any assistance at six weeks from start of illness. 16.9% reported being unable to work solely due to

**Data Availability Statement:** The survey data is available on request provided ethics committee approval for sharing the anonymised data is

granted. To request access conditional on approval, please email rgoinfo@soton.ac.uk.

**Funding:** The author(s) received no specific funding for this work.

**Competing interests:** The authors have no potentially competing interests to declare.

COVID-19 illness. 37.0% reported loss of income due to illness, and 64.4% said they were unable to perform usual activities/duties. Acute systems clustered broadly into two groups: a majority cluster (n = 2235, 88%) with cardiopulmonary predominant symptoms, and a minority cluster (n = 305, 12%) with multisystem symptoms. Similarly, ongoing symptoms broadly clustered in two groups; a majority cluster (n = 2243, 88.8%) exhibiting mainly cardiopulmonary, cognitive symptoms and exhaustion, and a minority cluster (n = 283, 11.2%) exhibiting more multisystem symptoms. Belonging to the more severe multisystem cluster was associated with more severe functional impact, lower income, younger age, being female, worse baseline health, and inadequate rest in the first two weeks of the illness, with no major differences in the cluster patterns when restricting analysis to the lab-confirmed subgroup.

## Conclusion

This is an exploratory survey of Long Covid characteristics. Whilst this is a non-representative population sample, it highlights the heterogeneity of persistent symptoms, and the significant functional impact of prolonged illness following confirmed or suspected SARS-CoV-2 infection. To study prevalence, predictors and prognosis, research is needed in a representative population sample using standardised case definitions.

## Introduction

The morbidity burden of the COVID-19 pandemic is becoming increasingly apparent and concerning. Long Covid describes the condition of not recovering for many weeks or months following acute SARS-CoV-2 infection [1]. It was first described and named as an umbrella term through a social media movement in Spring 2020 when many people with suspected or confirmed COVID-19 infection were not recovering weeks after onset of symptoms [2, 3]. Long Covid can occur regardless of the severity of the initial infection [4, 5]. The mechanisms underlying it are still largely unknown [6] and therefore it is premature to label all of its manifestations as a post viral illness [3]. Evidence describing the condition is scarce, but is starting to emerge on the long-term health impairment and organ damage following COVID-19 [7–11]. Patients are struggling to access adequate recognition, support, medical assessment and treatment for their condition, particularly those with no lab evidence of their infection during the first wave of the pandemic when testing was not accessible to those not hospitalised in the initial phase of their COVID-19 disease [12, 13].

The prevalence of Long Covid is still uncertain, but evidence is emerging that it is relatively common. Data from the UK's Office for National Statistics (ONS), based on a nationally representative non-institutionalised sample of lab-confirmed COVID-19 cases including asymptomatic ones, estimate a prevalence of 11.7% at 12 weeks from testing positive, increasing to 17.7% when considering only those symptomatic at the acute phase of the illness [14]. However, the detailed range of symptoms, disability, progression from the acute illness, and impact on work and daily activities are still not well described in such non-hospitalised population-based surveys. For example, the ONS study base their estimates on a list of 12 symptoms included in the ONS infection survey [14, 15], with some of the common symptoms of Long Covid such as chest pain, palpitations and cognitive problems missing from that list. Other

studies, some with a wider symptom list, estimate the prevalence of persisting symptoms to be higher at around one in three people for up to 18 weeks post infection [4, 5, 16].

There are more studies following-up hospitalised than non-hospitalised COVID-19 patients, with the assumption that hospitalisation indicates severe disease in most settings [17–19]. The natural history and pathology in those acutely severely ill with COVID-19 may be different to those developing Long Covid, but certain inflammatory or immunological mechanisms may be shared [20, 21]. The multisystem nature of the illness is a common feature. A multi-country web-based survey of suspected and confirmed COVID-19 cases found a range of 205 symptoms, with respondents who had a duration of illness over 6 months experiencing an average of 14 symptoms [10].

A rapid living systematic review concluded that there is currently insufficient evidence to provide a precise definition of Long Covid symptoms and prevalence [22]. The National Institute of Health and Care Excellence (NICE) has defined "post-COVID-19 syndrome" as signs or symptoms that develop during or after acute COVID-19, continue for more than 12 weeks and are not explained by an alternative diagnosis [23]. However, the 'signs or symptoms' that qualify for the definition are not specified. This may result in variation in diagnosis and referral among different clinicians, leading to inequalities in recognition and accessing services [24]. Many of those infected in spring 2020 did not have access to testing and therefore have struggled to receive recognition, diagnosis and support [12, 13]. This study was conceived following conversations with people with Long Covid in the community who perceived a lack of data on COVID-19 sequelae in non-hospitalised individuals and felt a need for their experience to be explored and documented.

In adults who self-reported Long Covid after suspected or confirmed COVID-19 and were not hospitalised in the first two weeks of their COVID-19 illness, we aimed to:

- Characterise the initial and the ongoing symptoms of Long Covid in terms of their range, nature, pattern, progression and what triggers and relieves them

- Describe the impact of Long Covid on daily activities and work

## Methods

This is a cross-sectional online survey using a convenience non-probability sampling method. The survey was posted by the study authors on social media websites (Twitter and Facebook), including on the Facebook Long Covid Support Group (membership at the time of posting was around 30,000, the group was founded in the UK but has international membership too), and the smaller UK doctors #longcovid Facebook Group. Subsequently, it was shared on the Survivor Corps Facebook Group (USA), and the Body Politic Support Group on Slack (international) by members of these groups. These social media groups were selected for posting the survey because we aimed to recruit people who identify themselves as living with Long Covid as well those who believe they have recovered from the illness. CH is the founder of the Facebook Long Covid Support Group, and MEO is on the administrative team for that Group. They both have experience of Long Covid.

The survey was available online in Microsoft Forms format, and open to complete for a period of one week, from November 7th to 14th 2020. The survey was only available in English, but responses were invited internationally, and not restricted to the UK, from those able to access the survey through social media and who fulfilled the inclusion criteria. The social media post contained brief information about the study, eligibility criteria and a link to the questionnaire. On opening the link, participants were taken to an in-depth participant information sheet. Participants gave their consent by answering 'yes' to a consent question.

Ethical approval was granted by the University of Southampton Faculty of Medicine Ethics Committee (ID 61434). Participants provided informed consent which was recorded digitally on the survey platform (Microsoft Forms). Participants had to consent to participating in the survey before they could access the questionnaire. Survey responses were anonymous, but participants who were willing to be contacted in the future for a follow-up survey were asked to consent to future contact and then provide contact details.

### Eligibility criteria

The survey was restricted to adults aged 18 years or over who thought they had COVID-19 (confirmed or suspected) and who were not hospitalised for the treatment of COVID-19 in the first two weeks of experiencing COVID-19 symptoms. The screening questions for the survey were the following.

- Are you aged 18 years or over?

- Do you think you have had COVID-19?

- Were you admitted to hospital in the first two weeks of experiencing COVID-19 symptoms?

If the participant answered 'no' to the first two questions or 'yes' to the third question, they could not progress further in the survey. Our survey provided an opportunity for people who were infected with SARS-CoV-2 but had not been hospitalised to participate in research to characterise their condition, since there were other studies following up hospitalised COVID-19 patients. In the UK, community testing for COVID-19 stopped on the 12[th] of March 2020 [25], and was not available throughout Spring 2020. Most of those who experienced COVID-19 symptoms and did not require hospital admission during that period did not have a positive test result. Therefore, the survey was open to those who did not have lab confirmation of their infection, but they had suspected or clinically diagnosed COVID-19. The survey was also open to people who had fully recovered from confirmed or suspected acute COVID-19.

### Questionnaire components

The questionnaire was co-produced working with public contributors experiencing Long Covid (CH and MEO). NAA also experienced Long Covid symptoms. Public contributor members of the COVID-19 Research Involvement Group (a Facebook group founded by MEO for the purpose of encouraging patient involvement in COVID research) gave feedback on early versions of the questionnaire to ensure that the questions were appropriate and relevant. The survey was amended according to their feedback. The questionnaire included questions primarily about the individual respondents and focused on minimising participant burden by collecting data deemed essential.

Questions included demographic information, baseline health, symptoms experienced at the start (first two weeks) of the COVID-19 illness (we refer to these as initial symptoms), the pattern of illness over the course, symptoms that remained/appeared over the longer term course (anytime after the first two weeks) of the illness (we refer to these as ongoing symptoms), functional status, impact on health, activity, ability to work including current employment status, and healthcare usage. We collected data on pre-existing health conditions as a binary (yes/no) variable and used an open text response to collect details on these conditions. We also asked if other members of the household had experienced symptoms of COVID-19 and the duration of their illness. With the exception of questions on initial symptoms and functional status at six weeks of illness, all questions captured responses at the time of survey completion.

The survey incorporated the Fatigue Severity Scale (FSS) to assess fatigue [26], and the Post-COVID-19 Functional Status (PCFS) Scale to assess functional status at six weeks from start of infection [27]. FSS consists of nine items scored on a seven-point Likert-type scale ranging from strongly disagree (1) to strongly agree (7). The nine items are combined into a total score calculated as the average of the individual item responses. A higher score indicates greater fatigue severity. We considered a score of 4 or above to indicate beyond normal levels of fatigue [28]. A PCFS scale variable was constructed consisting of grades 0–4 assigned based on yes/no responses to four component questions. Grade 0 reflects the absence of any functional limitation; grade 1 reflects the presence of symptoms, pain or anxiety without effect on activities (negligible functional limitations); grade 2 reflects the presence of symptoms, pain or anxiety requiring lower intensity of activities (slight functional limitations); grade 3 reflects the inability to perform certain activities (moderate functional limitations); and grade 4 reflects requiring assistance with activities of daily living (severe functional limitations).

## Statistical analysis

Data were downloaded from Microsoft Forms once the survey was taken offline. Statistical analysis was undertaken using Stata 15.0 and R. R packages used included readstata13, mclust, stats, and ggplot2. A minimum duration of illness of four weeks was defined as Long Covid for the purposes of this analysis. Confirmed infection was defined as reported positive result of nucleic acid amplification test (NAAT) such as PCR, and/or antibody test. Descriptive percentages and summary statistics were generated for the full sample and stratified separately for those with lab-confirmed and suspected infection. Univariate comparisons between those with and without confirmed COVID-19 infection were carried out using t-test or Mann Whitney U for continuous variables and chi square test for categorical variables. Univariate comparisons between those who tested positive, tested negative or were not tested for COVID-19 infection were carried out using ANOVA or Kruskal-Wallis test for continuous variables and chi square test for categorical variables. Complete case analysis was carried out as missing data was minimal.

Questions on initial and ongoing symptoms were used to categorise symptoms as not experienced, initial only (experienced in the first two weeks of the illness), new symptom developed after the acute phase, and initial symptom that remained as an ongoing symptom. Brain fog, poor concentration, memory problems and confusion are presented as distinct symptoms but were also used to derive a combined variable for "cognitive dysfunction". Similarly, chest pressure and chest tightness are distinct symptom questions used to derive a combined "chest pressure and/or tightness" variable. These derived variables were defined as having one or more of the component symptoms as initial and/or ongoing symptoms and categorised specifically for the derived variable. The percentages do not directly reflect the individual percentages of the component symptoms due to individuals having reported developing one (or more) component symptoms during different phases of the illness changing the distribution of the combined variable compared to the individual component symptoms. Ongoing symptoms were also categorised into the organ system affected (gastrointestinal, cardiopulmonary, neurological, systemic, nose/throat, pain and skin) (S1 Table).

## Clustering

We examined symptom clusters based on acute symptoms reported to have been experienced in the first two weeks of the illness, as well as with reported ongoing symptoms. We carried out hierarchical agglomerative clustering using hclust implemented in the R package stats using the complete method of clustering. We first generated a dissimilarity matrix based on

categorical binary data separately on symptoms during acute infection and with ongoing symptoms using Gowers distance. We used the silhouette method to identify the optimal number of clusters, by assessing both statistics for clusters 2 through 20. We examined the frequency of symptoms across different clusters in order to determine the clinical syndromes represented by the cluster. We examined patterns of transition of participants from acute clusters to ongoing clusters over time.

We also examined demographic, socioeconomic, and functional correlates of the ongoing symptom clusters. Categorical variables were initially analysed using the Chi square test. Means for continuous variables were compared by regressing the variable on cluster number, using the lm() function in R, in univariate analysis. We then examined predictors of cluster membership by using multiple logistic regression with cluster number as the dependent variable, and age, gender, ethnicity, income, education, alcohol consumption, smoking, baseline health, laboratory confirmation, acute symptom cluster membership, numbers of organ systems with at least one associated symptom, rest in the first two weeks of the illness, and pre-existing conditions as predictors. In order to account for the impact of duration of illness, and time-specific effects, we also included the month of infection as indicator variable to allow for heterogeneity of effect, and the reported duration of illness as covariate. Age category was also included as an indicator variable rather than an ordinal variable to allow for heterogeneity of effect.

As the full analysis included those with and without lab-confirmed diagnosis of Covid-19, we examined whether this was a significant predictor of cluster membership to assess whether clusters correlated with having lab-confirmation of infection. We also carried out an additional sensitivity analysis by clustering only those with lab confirmation to see if clusters obtained were different from the full sample analysis.

## Results

A total of 2644 participants completed the survey; 94 with reported length of illness of less than four weeks (n = 41) and those who had recovered from short acute COVID-19 (n = 53) were excluded. The numbers of individuals who had recovered from short acute or Long Covid were too small to enable comparison. 2550 participants were included in this analysis, of which 675 participants (26.5%) reported that they had SARS-CoV-2 infection confirmed through PCR and/or antibody tests. The mean duration of illness (experiencing symptoms) was 7.2 months (standard deviation (SD) 1.8 months, median 7.6 months, interquartile range 7.1–7.9), with a mean duration of 6.2 months (SD 2.4) in those lab-confirmed compared to 7.6 months (SD 1.3) in those who were not.

The mean age of participants was 46.5 years (SD 11 years). 82.8% were female and 93.3% were of White ethnicity. Responses were received from a range of places across the world, with the majority from the UK (79.9%: England 66.0%, Scotland 8.5%, Wales 4.5%, Northern Ireland 0.9%), North America (9.2%) and Europe (8.3%). The proportion of participants outside the UK was higher among those with lab-confirmed infection (29.1%) than among those with suspected infection (17.0%). In terms of educational attainment, 77.2% were qualified at university degree level or above (Table 1). Nineteen percent of participants reported that at least one other household member was also experiencing Long Covid (ill for 4 weeks or longer).

### Previous health

A small proportion of participants reported poor (1.3%) or fair (9.2%) health prior to COVID-19 infection, with 89.6% reporting good, very good or excellent health before COVID-19. 47.3% reported having pre-existing health conditions with asthma, hypertension, and

**Table 1. Demographics and baseline health of survey participants.**

| | Full sample | | Tested positive | | Tested negative or not tested | | p-value[a] |
|---|---|---|---|---|---|---|---|
| | n | % | n | % | n | % | |
| **Total n** | 2550 | | 675 | | 1793 | | |
| **Age, years (mean ± SD) (n = 2543)** | 46.5 ± 11.0 | | 45.3 ± 10.9 | | 46.6 ± 10.8 | | 0.01 |
| **Age, categorised** | | | | | | | |
| 18–30 | 189 | 7.4 | 68 | 10.1 | 118 | 6.6 | 0.006 |
| 31–45 | 997 | 39.2 | 271 | 40.2 | 712 | 39.8 | |
| 46–59 | 1051 | 41.4 | 275 | 40.8 | 741 | 41.5 | |
| ≥60 | 305 | 12.0 | 60 | 8.9 | 216 | 12.1 | |
| **Gender (n = 2547)** | | | | | | | |
| Male | 413 | 16.2 | 101 | 15.0 | 290 | 16.2 | 0.22 |
| Female | 2108 | 82.8 | 572 | 84.7 | 1477 | 82.5 | |
| Non-binary | 21 | 0.8 | 1 | 0.2 | 20 | 1.1 | |
| Prefer not to say | 3 | 0.1 | 1 | 0.2 | 2 | 0.1 | |
| Other | 2 | 0.1 | - | - | 2 | 0.1 | |
| **Country (n = 2523)** | | | | | | | |
| UK—England | 1665 | 66.0 | 410 | 61.5 | 1203 | 67.5 | <0.001 |
| UK—Scotland | 215 | 8.5 | 34 | 5.1 | 176 | 9.9 | |
| UK—Wales | 114 | 4.5 | 23 | 3.4 | 85 | 4.8 | |
| UK—Northern Ireland | 22 | 0.9 | 6 | 0.9 | 15 | 0.8 | |
| Outside the UK | 507 | 20.1 | 194 | 29.1 | 302 | 17.0 | |
| Africa | 18 | 0.7 | 16 | 2.4 | 2 | 0.1 | |
| Australia and New Zealand | 15 | 0.6 | 7 | 1.0 | 8 | 0.4 | |
| Europe | 210 | 8.3 | 60 | 9.0 | 145 | 8.1 | |
| South/Central America and Caribbean | 10 | 0.4 | 5 | 0.7 | 5 | 0.3 | |
| North America | 232 | 9.2 | 93 | 13.9 | 133 | 7.5 | |
| Asia | 15 | 0.6 | 7 | 1.0 | 8 | 0.4 | |
| Middle East | 7 | 0.3 | 6 | 0.9 | 1 | 0.1 | |
| **Ethnicity (n = 2533)** | | | | | | | |
| White | 2362 | 93.3 | 607 | 90.3 | 1688 | 94.4 | <0.001 |
| Mixed/Multiple ethnic backgrounds | 67 | 2.7 | 18 | 2.7 | 47 | 2.6 | |
| Asian | 64 | 2.5 | 25 | 3.7 | 36 | 2.0 | |
| Black/African/Caribbean | 23 | 0.9 | 15 | 2.2 | 8 | 0.5 | |
| Other | 14 | 0.6 | 7 | 1.0 | 7 | 0.4 | |
| Prefer not to say | 3 | 0.1 | - | - | 3 | 0.2 | |
| **Educational attainment (n = 2527)** | | | | | | | |
| No formal qualifications | 37 | 1.5 | 11 | 1.7 | 24 | 1.3 | 0.76 |
| O levels or equivalent | 209 | 8.3 | 57 | 8.6 | 145 | 8.1 | |
| A levels or equivalent | 331 | 13.1 | 79 | 11.8 | 236 | 13.2 | |
| University degree or above | 1950 | 77.2 | 520 | 78.0 | 1381 | 77.3 | |
| **Smoking status (n = 2537)** | | | | | | | |
| Non-smoker | 1577 | 62.2 | 424 | 62.9 | 1103 | 61.7 | 0.67 |
| Ex-smoker | 692 | 27.3 | 184 | 27.3 | 489 | 27.3 | |
| Current smoker | 268 | 10.6 | 66 | 9.8 | 197 | 11.0 | |
| **Alcohol intake in the 12 months before COVID-19 (n = 2539)** | | | | | | | |
| Do not drink | 91 | 3.6 | 35 | 5.2 | 54 | 3.0 | 0.01 |
| Did not drink in the past year | 254 | 10.0 | 50 | 7.4 | 192 | 10.7 | |

*(Continued)*

**Table 1.** (Continued)

| | Full sample | | Tested positive | | Tested negative or not tested | | p-value[a] |
|---|---|---|---|---|---|---|---|
| | n | % | n | % | n | % | |
| <Once a month | 452 | 17.8 | 137 | 20.3 | 306 | 17.1 | |
| Once a month | 210 | 8.3 | 62 | 9.2 | 144 | 8.0 | |
| Few times a month | 514 | 20.2 | 135 | 20.0 | 368 | 20.6 | |
| 1–3 times a week | 708 | 27.9 | 186 | 27.6 | 498 | 27.8 | |
| 4–6 times a week | 245 | 9.7 | 55 | 8.2 | 182 | 10.2 | |
| Everyday | 65 | 2.6 | 14 | 2.1 | 47 | 2.6 | |
| **Baseline health before COVID-19 (n = 2540)** | | | | | | | |
| Poor | 32 | 1.3 | 3 | 0.5 | 27 | 1.5 | 0.07 |
| Fair | 233 | 9.2 | 53 | 7.9 | 172 | 9.6 | |
| Good | 675 | 26.6 | 199 | 29.5 | 462 | 25.8 | |
| Very good | 1050 | 41.3 | 277 | 41.1 | 749 | 41.8 | |
| Excellent | 550 | 21.7 | 142 | 21.1 | 382 | 21.3 | |
| **Pre-existing health conditions (n = 2541)** | | | | | | | |
| No | 1339 | 52.7 | 337 | 49.9 | 965 | 53.8 | 0.08 |
| Yes | 1202 | 47.3 | 338 | 50.1 | 828 | 46.2 | |

[a]Comparisons between those with and without lab-confirmed COVID-19 used t-test for continuous and chi square test for categorical variables.

hyperthyroidism being the most common conditions reported (S2 Table). There were no significant differences in these proportions between those with and without lab-confirmation of infection (Table 1).

## Course of illness

The most common initial symptoms (first two weeks of the illness) were exhaustion (75.9%), headache (65.5%), chest pressure and/or tightness (64.5%), shortness of breath (61.7%), cough (58.5%), muscle aches (55.2%), fever (51.1%) and chills (51.0%) (Table 2). A significantly higher proportion of participants in the lab-confirmed group reported loss/altered smell or taste, loss of appetite, memory problems, headache, nasal symptoms/sneezing, joint pain and muscle aches during the acute phase, whereas a higher proportion of those without lab-confirmation reported chest pain, pressure and/or tightness (Table 2). In terms of ongoing symptoms, the most common were exhaustion (72.6%), cognitive dysfunction (brain fog, poor concentration, memory problems, confusion) (69.2%), chest pressure and/or tightness (52.6%), shortness of breath (54.2%), headache (46.0%), muscle aches (44.6%) and palpitations (42.0%) (Table 3 and Fig 1), with proportions reporting these symptoms comparable in those with and without lab-confirmation. Significant differences in reported prevalence of ongoing symptoms in those with and without lab-confirmation include altered/loss of sense of smell or taste and brain fog which were higher in those with lab-confirmation than without, whereas abdominal pain, nausea, chest pain, chest tightness, chills, hoarse voice, sore throat, sneezing and pins and needles were lower in those with lab-confirmation than without. Mean scores for each item on the Fatigue Severity Scale ranged between 5.2 and 6 (maximum (most severe) score is 7). Using a score of ≥4, the frequency of fatigue among survey participants was 86% with no statistically significant difference between those with and without lab-confirmation (Table 3).

**Table 2.** Initial symptoms experienced at the start of COVID-19 illness (first two weeks).

| | Full sample | | Tested positive | | Tested negative or not tested | | p-value[a] |
|---|---|---|---|---|---|---|---|
| | n | % | n | % | n | % | |
| **n** | 2540 | | 675 | | 1793 | | |
| Fever | 1298 | 51.1 | 362 | 53.6 | 893 | 49.8 | 0.09 |
| Cough | 1485 | 58.5 | 399 | 59.1 | 1037 | 57.8 | 0.57 |
| Altered or loss of sense of smell | 922 | 36.3 | 410 | 60.7 | 487 | 27.2 | <0.001 |
| Altered or loss of sense of taste | 921 | 36.3 | 388 | 57.5 | 510 | 28.4 | <0.001 |
| Abdominal pain | 562 | 22.1 | 150 | 22.2 | 402 | 22.4 | 0.92 |
| Diarrhoea | 855 | 33.7 | 235 | 34.8 | 601 | 33.5 | 0.54 |
| Loss of appetite | 946 | 37.2 | 293 | 43.4 | 624 | 34.8 | <0.001 |
| Nausea | 642 | 25.3 | 176 | 26.1 | 451 | 25.2 | 0.64 |
| Vomiting | 148 | 5.8 | 47 | 7.0 | 99 | 5.5 | 0.17 |
| Cognitive dysfunction | 1168 | 46.0 | 315 | 46.7 | 822 | 45.8 | 0.72 |
| Brain fog | 797 | 31.4 | 226 | 33.5 | 550 | 30.7 | 0.18 |
| Confusion | 539 | 21.2 | 137 | 20.3 | 385 | 21.5 | 0.52 |
| Memory problems | 475 | 18.7 | 152 | 22.5 | 311 | 17.4 | 0.003 |
| Poor concentration | 730 | 28.7 | 198 | 29.3 | 516 | 28.8 | 0.79 |
| Depression | 187 | 7.4 | 57 | 8.4 | 126 | 7.0 | 0.23 |
| Chest pain | 991 | 39.0 | 239 | 35.4 | 728 | 40.6 | 0.02 |
| Chest pressure | 1314 | 51.7 | 323 | 47.9 | 967 | 53.9 | 0.007 |
| Chest tightness | 1379 | 54.3 | 338 | 50.1 | 1016 | 56.7 | 0.003 |
| Palpitations | 754 | 29.7 | 215 | 31.9 | 521 | 29.1 | 0.18 |
| Shortness of breath | 1566 | 61.7 | 405 | 60.0 | 1121 | 62.5 | 0.25 |
| Chills | 1296 | 51.0 | 359 | 53.2 | 910 | 50.8 | 0.28 |
| Dizziness | 1079 | 42.5 | 304 | 45.0 | 738 | 41.2 | 0.08 |
| Exhaustion | 1928 | 75.9 | 514 | 76.2 | 1367 | 76.2 | 0.96 |
| Headache | 1663 | 65.5 | 480 | 71.1 | 1138 | 63.5 | <0.001 |
| Hoarse voice | 653 | 25.7 | 156 | 23.1 | 482 | 26.9 | 0.06 |
| Nasal symptoms | 717 | 28.2 | 231 | 34.2 | 466 | 26.0 | <0.001 |
| Sore throat | 1161 | 45.7 | 291 | 43.1 | 837 | 46.7 | 0.11 |
| Sneezing | 242 | 9.5 | 85 | 12.6 | 148 | 8.3 | 0.001 |
| Tinnitus | 339 | 13.4 | 104 | 15.4 | 217 | 12.1 | 0.03 |
| Joint pain | 890 | 35.0 | 290 | 43.0 | 584 | 32.6 | <0.001 |
| Leg pain | 573 | 22.6 | 179 | 26.5 | 370 | 20.6 | 0.002 |
| Muscle aches | 1402 | 55.2 | 425 | 63.0 | 936 | 52.2 | <0.001 |
| Pins and needles | 388 | 15.3 | 109 | 16.2 | 263 | 14.7 | 0.36 |
| Skin rash | 289 | 11.4 | 81 | 12.0 | 198 | 11.0 | 0.50 |
| Sleep disturbance | 909 | 35.8 | 243 | 36.0 | 638 | 35.6 | 0.85 |
| Number of initial symptoms, mean ± SD, median (interquartile range) | 12 ± 6 | | 13 ± 6 | | 12 ± 6 | | <0.001 |
| | 11 (7 to 16) | | 12 (8 to 17) | | 11 (7 to 15) | | |

[a]Comparisons between those with and without lab-confirmation of COVID-19 used t-test or Mann Whitney U for continuous and chi square test for categorical variables.

Participants reported experiencing a mean of 12 (SD 6, median 11, IQR 7–16) initial symptoms and 10 (SD 6, median 9, IQR 5–14) ongoing symptoms. The most common initial symptoms that persisted past the acute phase were exhaustion (59.1%), shortness of breath (41.3%), chest pressure and/or tightness (40.5%), and headache (37.5%). At least one symptom of

**Table 3. Ongoing symptoms, fatigue severity and organ systems affected.**

| | Full sample | | Tested positive | | Tested negative or not tested | | p-value[a] |
|---|---|---|---|---|---|---|---|
| | n | % | n | % | n | % | |
| n | 2526 | | 675 | | 1792 | | |
| **Ongoing symptoms** | | | | | | | |
| Fever | 217 | 8.6 | 46 | 6.8 | 167 | 9.3 | 0.05 |
| Cough | 587 | 23.3 | 158 | 23.4 | 413 | 23.1 | 0.85 |
| Altered or loss of sense of smell | 358 | 14.2 | 165 | 24.4 | 183 | 10.2 | <0.001 |
| Altered or loss of sense of taste | 313 | 12.4 | 141 | 20.9 | 164 | 9.2 | <0.001 |
| Abdominal pain | 427 | 16.9 | 97 | 14.4 | 319 | 17.8 | 0.04 |
| Diarrhoea | 398 | 15.8 | 95 | 14.1 | 293 | 16.4 | 0.17 |
| Loss of appetite | 283 | 11.2 | 69 | 10.2 | 210 | 11.7 | 0.30 |
| Nausea | 412 | 16.3 | 90 | 13.3 | 315 | 17.6 | 0.01 |
| Vomiting | 46 | 1.8 | 9 | 1.3 | 37 | 2.1 | 0.23 |
| Anxiety | 715 | 28.3 | 213 | 31.6 | 493 | 27.5 | 0.05 |
| Cognitive dysfunction | 1747 | 69.2 | 480 | 71.1 | 1232 | 68.8 | 0.26 |
| Brain fog | 1490 | 59.0 | 427 | 63.3 | 1034 | 57.7 | 0.01 |
| Confusion | 520 | 20.6 | 145 | 21.5 | 363 | 20.3 | 0.50 |
| Memory problems | 1094 | 43.3 | 294 | 43.6 | 777 | 43.4 | 0.93 |
| Poor concentration | 1138 | 45.1 | 304 | 45.0 | 814 | 45.4 | 0.86 |
| Depression | 397 | 15.7 | 106 | 15.7 | 283 | 15.8 | 0.96 |
| Chest pain | 891 | 35.3 | 214 | 31.7 | 656 | 36.6 | 0.02 |
| Chest pressure | 970 | 38.4 | 263 | 39.0 | 695 | 38.8 | 0.94 |
| Chest tightness | 1023 | 40.5 | 247 | 36.6 | 752 | 42.0 | 0.02 |
| Palpitations | 1062 | 42.0 | 270 | 40.0 | 774 | 43.2 | 0.15 |
| Shortness of breath | 1370 | 54.2 | 370 | 54.8 | 977 | 54.5 | 0.90 |
| Chills | 373 | 14.8 | 77 | 11.4 | 286 | 16.0 | 0.004 |
| Dizziness | 980 | 38.8 | 256 | 37.9 | 703 | 39.2 | 0.55 |
| Exhaustion | 1834 | 72.6 | 494 | 73.2 | 1298 | 72.4 | 0.71 |
| Headache | 1161 | 46.0 | 320 | 47.4 | 827 | 46.2 | 0.56 |
| Hoarse voice | 453 | 17.9 | 103 | 15.3 | 342 | 19.1 | 0.03 |
| Nasal symptoms | 471 | 18.7 | 110 | 16.3 | 353 | 19.7 | 0.05 |
| Sore throat | 591 | 23.4 | 128 | 19.0 | 454 | 25.3 | 0.001 |
| Sneezing | 188 | 7.4 | 37 | 5.5 | 146 | 8.2 | 0.02 |
| Tinnitus | 662 | 26.2 | 159 | 23.6 | 477 | 26.6 | 0.12 |
| Joint pain | 950 | 37.6 | 252 | 37.3 | 681 | 38.0 | 0.76 |
| Leg pain | 668 | 26.4 | 184 | 27.3 | 472 | 26.3 | 0.65 |
| Muscle aches | 1126 | 44.6 | 303 | 44.9 | 795 | 44.4 | 0.82 |
| Pins and needles | 667 | 26.4 | 156 | 23.1 | 501 | 28.0 | 0.02 |
| Skin rash | 299 | 11.8 | 73 | 10.8 | 217 | 12.1 | 0.37 |
| Sleep disturbance | 952 | 37.7 | 241 | 35.7 | 691 | 38.6 | 0.19 |
| Number of ongoing symptoms, mean ± SD, median (interquartile range) | 10 ± 6 | | 10 ± 6 | | 10 ± 6 | | 0.49 |
| | 9 (5 to 14) | | 9 (5 to 13) | | 9 (5 to 14) | | |
| Fatigue Severity Scale score, mean ± SD (n = 2000) | 5.5 ± 1.4 | | 5.5 ± 1.4 | | 5.5 ± 1.4 | | 0.38 |
| Score ≥4% | 86 | | 84 | | 86 | | |
| **Number of organ systems affected** | | | | | | | |
| 1 | 121 | 4.8 | 29 | 4.3 | 91 | 5.1 | 0.02 |
| 2 | 253 | 10.0 | 81 | 12.0 | 164 | 9.2 | |

*(Continued)*

**Table 3.** (Continued)

| | Full sample | | Tested positive | | Tested negative or not tested | | p-value[a] |
|---|---|---|---|---|---|---|---|
| | n | % | n | % | n | % | |
| 3 | 437 | 17.3 | 120 | 17.8 | 308 | 17.2 | |
| 4 | 623 | 24.7 | 185 | 27.4 | 421 | 23.5 | |
| 5 | 551 | 21.8 | 145 | 21.5 | 393 | 21.9 | |
| 6 | 380 | 15.0 | 77 | 11.4 | 295 | 16.5 | |
| 7 | 119 | 4.7 | 29 | 4.3 | 88 | 4.9 | |
| **Organ systems affected by symptoms** | | | | | | | |
| Gastrointestinal | 909 | 36.0 | 220 | 32.6 | 666 | 37.2 | 0.04 |
| Chest (cardiopulmonary) | 2070 | 82.0 | 552 | 81.8 | 1471 | 82.1 | 0.86 |
| Neurological | 2164 | 85.7 | 582 | 86.2 | 1530 | 85.4 | 0.60 |
| Systemic | 2035 | 80.6 | 541 | 80.2 | 1445 | 80.6 | 0.79 |
| Nose/Throat | 1036 | 41.0 | 232 | 34.4 | 788 | 44.0 | <0.001 |
| Pain | 1785 | 70.7 | 481 | 71.3 | 1261 | 70.4 | 0.67 |
| Skin | 299 | 11.8 | 73 | 10.8 | 217 | 12.1 | 0.37 |

[a]Comparisons between those with and without lab-confirmation of COVID-19 used t-test or Mann Whitney U for continuous variables and chi square test for categorical variables.

cognitive dysfunction was present in the initial first two weeks and persisted throughout the illness in 36.9% of participants but was also reported as new symptom(s) after the acute phase of the illness in 32.3% of participants, including brain fog (36.1%), memory problems (30.7%), and poor concentration (27.4%) (S3 Table).

Ongoing symptoms affected three or more organ systems (gastrointestinal, cardiopulmonary, neurological, systemic, nose/throat, pain and skin) in 83.5% of participants, with 21.8% reporting symptoms that affected five systems, 15.0% six systems, and 4.7% seven systems (Table 3). Although a similar proportion reported ongoing symptoms that affected three or more organ systems, a higher proportion of those without lab-confirmation (43.3%) reported ongoing symptoms that affected five or more organ systems than those with lab-confirmation (37.2%). The majority of participants reported a course of illness that was fluctuating (intensity of symptoms changes but symptoms never completely go away) (57.7%) or symptoms 'coming and going'/relapsing (experience symptom-free periods in between relapses) (17.6%). 72.8% of participants experienced symptoms daily. Exhaustion improved on resting in 35.3% of participants. The majority of participants (60.4%) said that exertion (exercise/work) was not the only cause of exhaustion (Table 4) with no difference between those with and without lab-confirmation. Participants with lab-confirmation of infection were more likely to report they had rested well in the first two weeks of the illness (60.4% vs 51.8%).

Only 2.3% of participants reported that they felt they had recovered to baseline health before COVID-19 with a further 20.1% reporting that they were not symptomatic at the time of completing the survey but did not feel they had recovered to pre-infection health and/or activity levels. The remaining 77.7% reported that they were experiencing symptoms at the time of completing the survey (Table 4). The proportions reporting recovery and still experiencing symptoms were similar in those with and without lab confirmation of infection. Of those who reported completely recovering from Long Covid (n = 58), the duration of illness was 1–4 months for 65.5% and six months or longer for 13.8% (S4 Table).

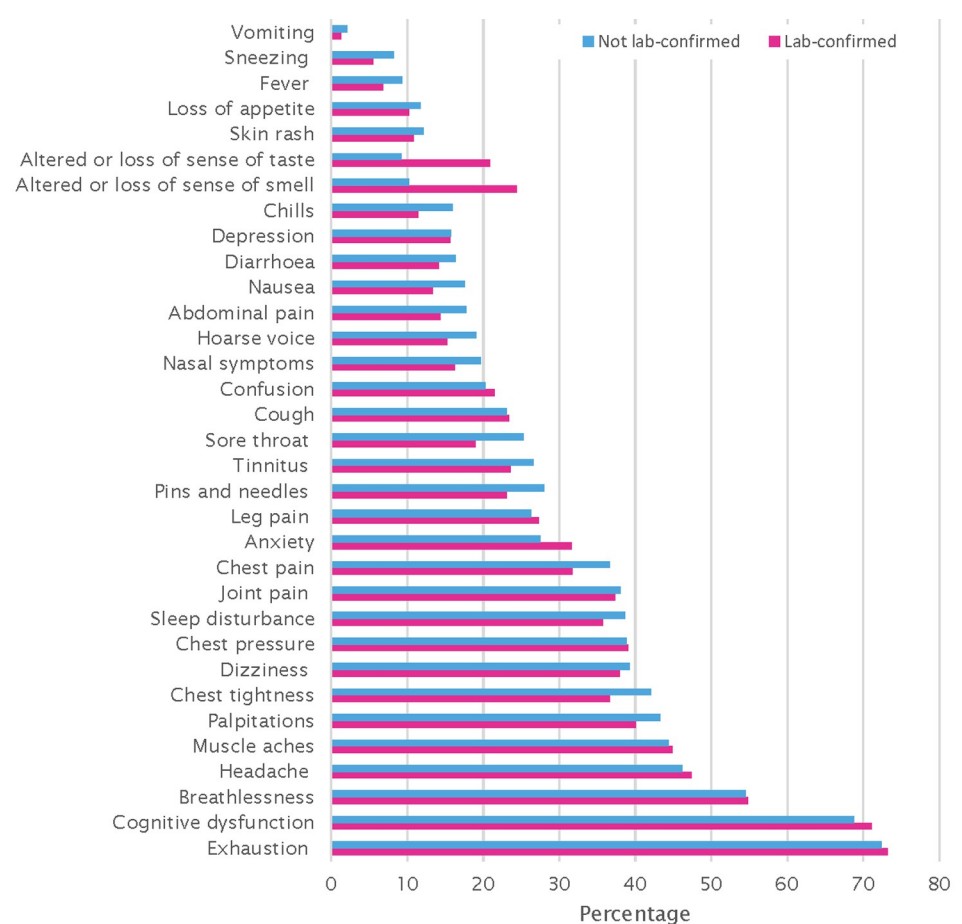

**Fig 1. Frequency of reported ongoing symptoms in survey participants (n = 2526).**

Common triggers that exacerbated existing symptoms or caused symptoms to return included physical activity (77.2%), stress (55.1%), disturbance in sleep patterns (46.9%), cognitive activity (42.2%), and domestic chores (35.0%). 23.2% reported symptoms varying by time of day. 15.8% of participants also reported not always being able to identify a trigger and sometimes symptoms returned or worsened without a trigger. Just over half of participants (54.3%) reported sufficient rest in the acute phase of the illness, with 26.0% reporting less rest than they would have liked due to caring or other responsibilities (Table 4). A higher proportion of participants with lab-confirmation (60.4%) than those without (51.8%) reported sufficient rest in the acute phase.

## Functional ability

At the time of completing the survey, being ill still affected respondents' ability to carry out domestic chores (84.3%), leisure (84.8%) and social (77.1%) activities, work (74.9%), self-care (50.0%), childcare (35.8%), and caring for other adults (26.1%), as well as affecting their mental health (63.7%). Using the PCFS Scale to describe how Long Covid affected daily activities at six weeks from the start of symptoms, nearly a third (32.3%) reported that they were unable to live alone without any assistance, and 34.5% reported moderate functional limitations (able to take care of self but not perform usual duties/activities). A higher proportion of participants without lab-confirmation reported moderate or severe functional limitations (68.1%) compared to

**Table 4. Duration, pattern and triggers of illness.**

| | Full sample | | Tested positive | | Tested negative or not tested | | p-value[a] |
|---|---|---|---|---|---|---|---|
| | Overall n | % | n | % | n | % | |
| **Total n** | 2550 | | 675 | | 1793 | | |
| **Well rested in first two weeks of illness (n = 2536)** | | | | | | | |
| No | 437 | 17.2 | 98 | 14.5 | 326 | 18.2 | 0.002 |
| Yes | 1376 | 54.3 | 407 | 60.4 | 928 | 51.8 | |
| Less than I would have liked | 658 | 26.0 | 154 | 22.9 | 489 | 27.3 | |
| Not sure | 65 | 2.6 | 15 | 2.2 | 49 | 2.7 | |
| **Back to baseline health (n = 2538)** | | | | | | | |
| No, still symptomatic | 1971 | 77.7 | 531 | 78.7 | 1383 | 77.1 | 0.72 |
| No, but not symptomatic | 509 | 20.1 | 130 | 19.3 | 369 | 20.6 | |
| Yes | 58 | 2.3 | 14 | 2.1 | 41 | 2.3 | |
| **Duration of illness, weeks (mean ± SD) (n = 2458)** | 31.3 ± 7.8 | | 26.9 ± 10.6 | | 32.9 ± 5.7 | | <0.001 |
| **Duration of illness, months (mean ± SD) (n = 2458)** | 7.2 ± 1.8 | | 6.2 ± 2.4 | | 7.6 ± 1.3 | | <0.001 |
| **Pattern of illness (n = 2519)** | | | | | | | |
| Constant throughout | 146 | 5.8 | 43 | 6.4 | 97 | 5.4 | 0.26 |
| Gradually got worse | 273 | 10.8 | 69 | 10.2 | 201 | 11.3 | |
| Gradually got better | 201 | 8.0 | 63 | 9.3 | 130 | 7.3 | |
| Fluctuating | 1454 | 57.7 | 394 | 58.5 | 1033 | 57.8 | |
| Relapsing/Comes and goes | 445 | 17.6 | 105 | 15.5 | 325 | 18.2 | |
| **Symptom frequency (n = 2511)** | | | | | | | |
| Daily | 1827 | 72.8 | 485 | 72.2 | 1290 | 72.4 | 0.19 |
| >3 times a week | 425 | 16.9 | 126 | 18.8 | 295 | 16.6 | |
| Once a week | 95 | 3.8 | 24 | 3.6 | 70 | 3.9 | |
| Once a fortnight | 50 | 2.0 | 7 | 1.0 | 43 | 2.4 | |
| Once a month | 32 | 1.3 | 6 | 0.9 | 26 | 1.5 | |
| <Once a month | 20 | 0.8 | 9 | 1.3 | 11 | 0.6 | |
| Daily and reduced over time | 21 | 0.8 | 6 | 0.9 | 15 | 0.8 | |
| Episodic | 34 | 1.4 | 8 | 1.2 | 26 | 1.5 | |
| Variable | 7 | 0.3 | 1 | 0.2 | 6 | 0.3 | |
| **Triggers for return or exacerbation of symptoms (n = 2474)** | | | | | | | |
| Physical Activity | 1911 | 77.2 | 500 | 75.8 | 1364 | 77.6 | 0.33 |
| Diet | 454 | 18.4 | 90 | 13.6 | 351 | 20.0 | <0.001 |
| Hormonal | 582 | 23.5 | 138 | 20.9 | 438 | 24.9 | 0.04 |
| Cognitive activity | 1045 | 42.2 | 281 | 42.6 | 743 | 42.3 | 0.90 |
| Work | 705 | 28.5 | 206 | 31.2 | 485 | 27.6 | 0.08 |
| Social activity | 713 | 28.8 | 180 | 27.3 | 516 | 29.4 | 0.31 |
| Stress | 1364 | 55.1 | 332 | 50.3 | 994 | 56.6 | 0.006 |
| Time of day | 573 | 23.2 | 143 | 21.7 | 412 | 23.5 | 0.35 |
| Sleep disturbance | 1161 | 46.9 | 287 | 43.5 | 847 | 48.2 | 0.04 |
| Domestic chores | 866 | 35.0 | 213 | 32.3 | 632 | 36.0 | 0.09 |
| Caring responsibilities | 411 | 16.6 | 91 | 13.8 | 307 | 17.5 | 0.03 |
| Unknown | 404 | 15.8 | 123 | 18.2 | 275 | 15.3 | 0.08 |
| Other—Talking | 30 | 1.2 | 5 | 0.8 | 23 | 1.3 | 0.26 |
| Other—Posture | 18 | 0.7 | 2 | 0.3 | 15 | 0.8 | 0.15 |
| **Exhaustion improves on rest (n = 2332)** | | | | | | | |
| No | 317 | 13.6 | 99 | 15.7 | 208 | 12.6 | 0.26 |

*(Continued)*

**Table 4.** (Continued)

| | Full sample | | Tested positive | | Tested negative or not tested | | p-value[a] |
|---|---|---|---|---|---|---|---|
| | **Overall n** | **%** | **n** | **%** | **n** | **%** | |
| Yes | 823 | 35.3 | 222 | 35.2 | 582 | 35.4 | |
| Sometimes | 1192 | 51.1 | 310 | 49.1 | 855 | 52.0 | |
| **Exhaustion caused by exertion (exercise/work) only (n = 2343)** | | | | | | | |
| No | 1415 | 60.4 | 384 | 60.8 | 1006 | 60.8 | 0.96 |
| Yes | 241 | 10.3 | 62 | 9.8 | 174 | 10.5 | |
| Sometimes | 506 | 21.6 | 135 | 21.4 | 352 | 21.3 | |
| Do not know | 181 | 7.7 | 51 | 8.1 | 122 | 7.4 | |

[a]Comparisons between those with and without lab-confirmation of COVID-19 used t-test for continuous and chi square test for categorical variables.

those with lab-confirmation (61.2%). 89.5% of participants said they avoided certain activities/ duties at six weeks from onset of illness. Only 10.3% reported no or negligible functional limitations (Table 5).

## Work

At the time of responding to the survey, 9.7% reported working reduced hours, 19.1% reported being unable to work (out of which 88.3% was reported to be solely due to COVID-19 illness),

**Table 5. Functional ability of study participants.**

| | Full sample | | Tested positive | | Tested negative or not tested | | p-value[a] |
|---|---|---|---|---|---|---|---|
| | **n** | **%** | **n** | **%** | **n** | **%** | |
| **Total n** | 2550 | | 675 | | 1793 | | |
| **Post-COVID-19 Functional Status Scale components at 6 weeks from start of symptoms** | | | | | | | |
| Unable to live alone (n = 2499) | 808 | 32.3 | 187 | 28.0 | 599 | 33.8 | 0.006 |
| Unable to perform activities/duties (n = 2525) | 1627 | 64.4 | 397 | 58.8 | 1191 | 66.4 | <0.001 |
| Suffer from symptoms, depression, pain or anxiety (n = 2538) | 2521 | 99.3 | 670 | 99.3 | 1782 | 99.4 | 0.73 |
| Avoid activities/duties (n = 2486) | 2224 | 89.5 | 574 | 86.7 | 1602 | 90.4 | 0.008 |
| **Post-COVID-19 Functional Status Scale at 6 weeks from start of symptoms (n = 2498)** | | | | | | | 0.01 |
| No functional limitations | 17 | 0.7 | 5 | 0.7 | 11 | 0.6 | |
| Negligible functional limitations | 242 | 9.6 | 79 | 11.7 | 158 | 8.8 | |
| Slight functional limitations | 588 | 23.3 | 178 | 26.4 | 403 | 22.5 | |
| Moderate functional limitations | 871 | 34.5 | 226 | 33.5 | 622 | 34.7 | |
| Severe functional limitations | 808 | 32.0 | 187 | 27.7 | 599 | 33.4 | |
| **At the time of survey completion, being ill affected (n = 2478):** | | | | | | | |
| Self-care | 1238 | 50.0 | 282 | 42.3 | 928 | 52.5 | <0.001 |
| Childcare | 887 | 35.8 | 221 | 33.2 | 650 | 36.8 | 0.10 |
| Caring for other adults | 646 | 26.1 | 166 | 24.9 | 461 | 26.1 | 0.56 |
| Domestic chores | 2088 | 84.3 | 531 | 79.7 | 1517 | 85.8 | <0.001 |
| Work | 1857 | 74.9 | 517 | 77.6 | 1324 | 74.9 | 0.16 |
| Leisure activities | 2101 | 84.8 | 537 | 80.6 | 1525 | 86.3 | <0.001 |
| Social activities | 1911 | 77.1 | 491 | 73.7 | 1383 | 78.2 | 0.02 |
| Mental health | 1579 | 63.7 | 433 | 65.0 | 1122 | 63.5 | 0.48 |

[a]Comparisons between those with and without lab-confirmation of COVID-19 used chi square test for categorical variables.

Table 6. Employment status and impact of illness on work.

| | Full sample | | Tested positive | | Tested negative/not tested | | p-value[a]* |
|---|---|---|---|---|---|---|---|
| | n | % | n | % | n | % | |
| **Total n** | 2550 | | 675 | | 1793 | | |
| **Employment status at time of survey completion (n = 2507)** | | | | | | | |
| Working full-time | 919 | 36.7 | 306 | 45.3 | 606 | 33.8 | <0.001 |
| Working part-time | 340 | 13.6 | 72 | 10.7 | 265 | 14.8 | |
| Furloughed | 58 | 2.3 | 9 | 1.3 | 47 | 2.6 | |
| Working reduced hours | 243 | 9.7 | 65 | 9.6 | 176 | 9.8 | |
| Unemployed/Looking for work | 45 | 1.8 | 9 | 1.3 | 36 | 2.0 | |
| Unpaid (Volunteer, Carer) | 14 | 0.6 | 3 | 0.4 | 11 | 0.6 | |
| Student | 61 | 2.4 | 20 | 3.0 | 41 | 2.3 | |
| Homemaker | 101 | 4.0 | 16 | 2.4 | 79 | 4.4 | |
| Unable to work | 478 | 19.1 | 133 | 19.7 | 342 | 19.1 | |
| Made redundant/took early retirement | 47 | 1.9 | 6 | 0.9 | 39 | 2.2 | |
| Retired | 155 | 6.2 | 27 | 4.0 | 115 | 6.4 | |
| Off sick | 46 | 1.8 | 9 | 1.3 | 36 | 2.0 | |
| **Lost job or had/chose to stop work (n = 2483)** | | | | | | | |
| No | 1947 | 78.4 | 562 | 83.9 | 1362 | 76.4 | <0.001 |
| No but was furloughed | 165 | 6.7 | 25 | 3.7 | 136 | 7.6 | |
| Yes | 371 | 14.9 | 83 | 12.4 | 284 | 15.9 | |
| **Had time off sick (n = 2484)** | | | | | | | |
| No | 709 | 28.5 | 171 | 25.3 | 535 | 29.8 | <0.001 |
| No but was furloughed | 126 | 5.1 | 20 | 3.0 | 105 | 5.9 | |
| Yes | 1649 | 66.4 | 484 | 71.7 | 1153 | 64.3 | |
| **Time off sick, days (median, IQR) (n = 1564)** | 60 | | 54 | | 60 | | 0.56 |
| | 20 to 129 | | 22 to 129 | | 20 to 129 | | |
| **Loss of income due to COVID-19 illness (n = 2479)** | | | | | | | |
| No | 1548 | 62.4 | 450 | 66.7 | 1092 | 60.9 | 0.008 |
| Yes | 931 | 37.6 | 225 | 33.3 | 701 | 39.1 | |
| **Days income lost/too ill to work (median, IQR) (n = 622)** | 120 | | 84 | | 129 | | <0.001 |
| | 50 to 172 | | 30 to 151 | | 60 to 172 | | |

[a]Comparisons between those with and without lab-confirmation of COVID-19 used t-test or Mann Whitney U for continuous and chi square test for categorical variables.

and 1.9% reported being made redundant or having taken early retirement (Table 6 and Fig 2). The most common reported reason for working reduced hours was COVID-19 illness (96.5%). Those with lab confirmation of infection were more likely to be working full-time (45.3%) at the time of responding to the survey than those who were not tested or tested negative (33.8%). 66.4% reported taking time off sick and 5.1% reported not needing to take time off sick as they were furloughed. 71.7% of those with lab-confirmation reported taking time off sick compared to 64.3% of those without lab-confirmation. The median time off sick was 60 (IQR 20 to 129) days. 37.6% reported a loss of income due to illness (median reported number of days for which income is lost 120, IQR 50 to 172). This was significantly higher for those with no lab confirmation (median 129, IQR 60 to 172) compared to those with lab confirmation (median 84, IQR 30 to 151).

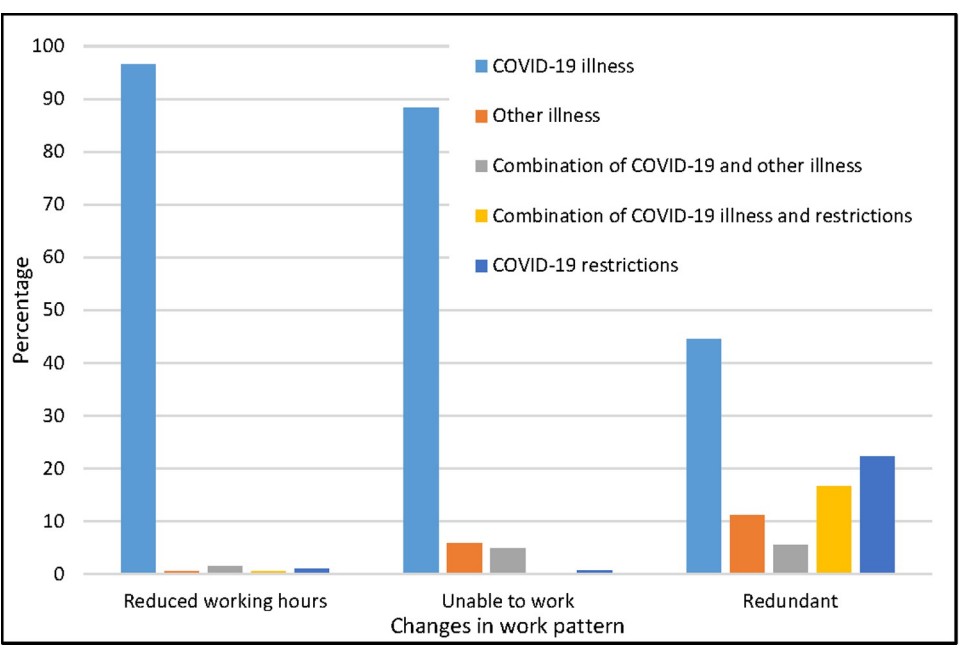

**Fig 2. Reasons for change in work pattern in those reporting reduced work hours (n = 243), being unable to work (n = 478) or being made redundant/taking early retirement (n-47) (total n = 768).**

## Healthcare utilisation

Most participants reported at least one or more type of healthcare service usage (GP, calls to non-emergency medical care number, accident and emergency department, hospital outpatient appointments) with 12% admitted to hospital after 2 weeks from onset of illness.

## Lab confirmation of infection

Out of the 2550 participants, 675 (26.5%) reported lab confirmation of infection by either PCR or antibody test and 82 did not answer the question on testing for lab confirmation of infection (3.2%) (27.4% lab-confirmed out of n = 2468 who answered the testing questions). 1582 participants (62%) reported having a PCR test with 426 testing positive (27% of those tested). The date of PCR test was available for 1491 of these 1582 participants. Twenty percent (n = 304) were first tested at 0–5 days of onset of symptoms, with 72.7% of this group testing positive, while 80% (n = 1187) were first tested $\geq$6 days of onset of symptoms, with 12.6% of this group testing positive.

1172 participants (46%) reported having an antibody test with 369 testing positive (31% of those tested). The date of antibody testing was available for 1120 of 1172 participants who reported having an antibody test. 26.3% (n = 294) were first tested between 2 to 12 weeks of onset of symptoms and 42.1% of them tested positive, while 72.5% (n = 812) were first tested $\geq$12 weeks from onset of symptoms with 27.7% testing positive. 820 participants (32%) reported having both a PCR and antibody test of which 120 (15%) tested positive for both, 48 (6%) positive for PCR only and 122 (15%) positive for antibodies only. Overall, 5% (n = 120) tested positive for both PCR and antibodies. Out of the 168 participants who tested positive for PCR and had an antibody test, 29% tested negative for antibodies. Out of the 652 participants who tested negative for PCR and had an antibody test, 19% tested positive for antibodies (S1 Fig). All the demographics, initial and ongoing symptoms are presented by the three categories of positive, negative and not tested in the (S5–S7 Tables).

## Clustering

Thirty-four symptoms were used in clustering for acute symptoms (Table 2) and 35 for ongoing symptoms (Table 3). Clustering based on acute symptoms (initial symptoms experienced during the first two weeks) identified two clusters as the optimal number of clusters (S2 Fig). Acute symptom cluster (ASC) 1 consists of the majority of participants (88%, n = 2235) who exhibit predominantly cardiopulmonary symptoms (cough, shortness of breath, chest pressure/tightness, chest pain) and exhaustion, while ASC2 consists of the remaining 12% (n = 305) who exhibit multisystem symptoms (S3 Fig). The most common acute symptoms in ASC2 include shortness of breath, chest pressure/tightness, chest pain, palpitations, cough (cardiopulmonary); appetite loss, diarrhoea (gastrointestinal); poor concentration, dizziness, brain fog, confusion (neuro-cognitive); sore throat, hoarse voice (nose/throat); headache, muscle ache, joint pain (pain); and exhaustion, chills, sleep disturbance and fever (systemic). On examining ongoing symptoms among ASCs 1 and 2, we found that although the differences between the groups persisted, they became less distinct primarily due to a large proportion of participants in ASC1 developing ongoing symptoms of cognitive dysfunction in addition to the predominantly cardiopulmonary symptoms over time.

On clustering participants based on ongoing symptoms, we once again identified two optimal clusters (Fig 3), with ongoing symptom cluster (OSC) 1 predominantly including participants with cardiopulmonary symptoms (shortness of breath, chest pain, chest pressure/tightness, palpitations), neuro-cognitive symptoms (brain fog, poor concentration, memory problems, dizziness), and exhaustion (n = 2243, 88.8%); and OSC2, a minority cluster, including multisystem ongoing symptoms (n = 283, 11.2%). The most common ongoing symptoms in OSC2 include shortness of breath, chest pain, chest pressure/tightness, palpitations (cardiopulmonary); brain fog, poor concentration, memory problems, dizziness, confusion, pins and needles (neuro-cognitive); sore throat, hoarse voice, nasal symptoms (nose/throat); headache, joint pain, leg pain, muscle ache (pain); and exhaustion, sleep disturbance and chills (systemic). In univariate analysis, membership of OCS2 was associated with worse fatigue (FSS)

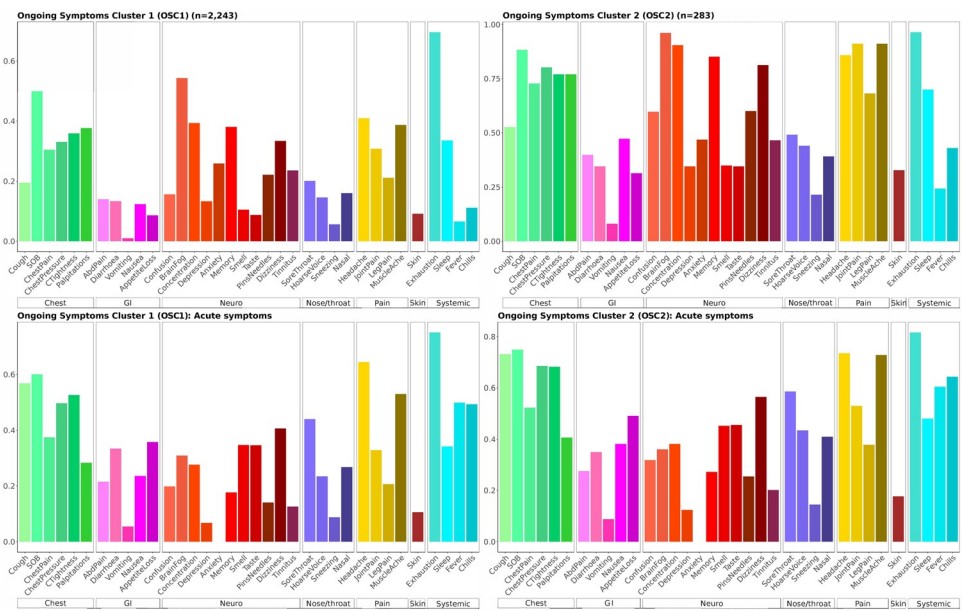

**Fig 3. Two clusters of ongoing symptoms and acute symptoms among these clusters.**

and PCFS scores; needing to take time off sick; compromised ability to carry out self-care, domestic chores, care for other adults and childcare, work, participate in leisure, or social activities; greater risk of losing employment or needing to stop work; and loss of income. Membership of OSC2 was also associated with having a pre-existing condition, poorer baseline health, and greater healthcare usage with a higher number of GP consultations (Table 7).

**Table 7. Correlates of Ongoing symptom clusters (n = 2526).**

| Characteristic[a] | Ongoing symptom cluster (OSC) 1 | Ongoing symptom cluster (OSC) 2 | p-value[b] |
|---|---|---|---|
| Age (mean) | 46.60 | 44.87 | 0.01 |
| Gender (Male) | 17 | 7 | <0.001 |
| Fatigue Severity Scale Score | 5.36 | 6.30 | <0.001 |
| Post COVID-19 Functional Scale Score (PCFS) | 2.82 | 3.29 | <0.001 |
| Duration of illness, days | 219.73 | 221.29 | 0.65 |
| Number of A&E visits | 0.74 | 0.90 | 0.07 |
| Number of GP consultations | 4.71 | 6.10 | <0.001 |
| Number of hospital out-patient appointments | 1.74 | 2.08 | 0.03 |
| Number of days off sick | 74.84 | 88.77 | 0.004 |
| Number of days of income lost | 109.97 | 136.88 | <0.001 |
| Pre-existing health conditions (Yes) | 46 | 60 | <0.001 |
| **Alcohol consumption in 12 months before COVID-19 infection** | | | <0.001 |
| Do not drink | 3 | 5 | |
| Did not drink in the past year | 10 | 11 | |
| <Once a month | 17 | 26 | |
| Once a month | 8 | 9 | |
| Few times a month | 20 | 22 | |
| 1–3 times a week | 29 | 20 | |
| 4–6 times a week | 10 | 7 | |
| Everyday | 3 | 1 | |
| **Self-reported health before COVID-19 infection** | | | <0.001 |
| Poor | 1 | 4 | |
| Fair | 9 | 15 | |
| Good | 26 | 33 | |
| Very good | 42 | 35 | |
| Excellent | 23 | 14 | |
| **Being ill affected** | | | |
| Self-care | 47 | 75 | <0.001 |
| Childcare | 34 | 43 | 0.002 |
| Caring for other adults | 24 | 42 | <0.001 |
| Domestic chores | 83 | 97 | <0.001 |
| Work | 74 | 84 | <0.001 |
| Leisure activities | 84 | 94 | <0.001 |
| Social activities | 75 | 91 | <0.001 |
| Mental health | 62 | 79 | <0.001 |
| **Hospitalisation for treatment of Long Covid symptoms** | | | <0.001 |
| No | 88 | 83 | |
| Ward–day stay | 5 | 9 | |
| Ward–overnight stay | 7 | 5 | |
| High dependency unit | 0 | 2 | |
| Intensive care unit | 0 | 0 | |

*(Continued)*

**Table 7.** (Continued)

| Characteristic[a] | Ongoing symptom cluster (OSC) 1 | Ongoing symptom cluster (OSC) 2 | p-value[b] |
|---|---|---|---|
| Ambulatory care | 0 | 0 | |
| **Lost job or had/chose to stop work** | | | <0.001 |
| No | 80 | 69 | |
| No but was furloughed | 6 | 8 | |
| Yes | 14 | 23 | |
| **Had time off sick** | | | <0.001 |
| No | 30 | 18 | |
| No but was furloughed | 5 | 5 | |
| Yes | 65 | 77 | |
| **Loss of income due to COVID-19 illness** | | | <0.001 |
| No | 64 | 48 | |
| Yes | 36 | 52 | |

[a]Summary statistics are expressed as means for continuous variables and percentages for categorical variables.

[b]Categorical variables were compared using the chi2 test and continuous variables were compared by regressing the variable on cluster number.

Multivariate fully adjusted analysis showed that being female (OR = 2.0, 95% confidence interval (CI) 1.2, 3.4), poor baseline health (OR = 3.4, 95% CI 1.2, 9.8), being a member of ACS2 (OR = 2.5, 95% CI 1.7, 3.5), a higher number of acute symptoms related to different organ systems (OR = 1.2, 95% CI 1.04, 1.31) were positively associated with membership of the more severe OCS2 cluster. Older age (>60 years) (OR = 0.35, 95% CI 0.19, 0.66), higher income (OR = 0.85 per increase in income category, 95% CI 0.75, 0.95), and sufficient rest in the first two weeks of the illness (OR = 0.68, 95% CI 0.46, 0.99) seemed to be protective against OCS2. OSC2 membership was not related to the duration since onset of acute symptoms (Fig 4).

In sensitivity analyses, we restricted to only participants who reported lab confirmation of infection. On hierarchical clustering into two clusters, consistent with our clustering on the whole dataset, we once again identified a majority cluster with cardiopulmonary, neurocognitive symptoms, and exhaustion dominating (n = 576), and a minority multisystem cluster where symptoms related to all systems were common (n = 99) (S4 Fig). We found high correlation between ongoing clusters identified with the whole dataset, and those identified when limiting data to only those with lab confirmation (r = 0.56, p<0.001).

## Transition between clusters

On examining the membership of acute symptom clusters by number of systems with at least one symptom, we found that 98% of those in ASC2 had 5 or more systems involved compared with 56% in ASC1. Even though acute symptom clustering strongly predicts ongoing symptom clusters, there is movement between clusters. Of those in OSC2, 73% had 5 or more systems involved compared with 59% in the OSC1. Both clusters had more multisystem involvement during the acute infection phase than the ongoing symptoms phase. Among 2223 participants clustering in ASC1, 9% (n = 202) move into OSC2 over time, suggesting increase in severity. Among 305 participants in ASC2, 27% (n = 81) remain in this cluster, with the remaining moving into OSC1, with cardiopulmonary, neurological, and fatigue symptoms predominating. Movement from ASC1 into OSC2 appears to be dependent on the number of organ

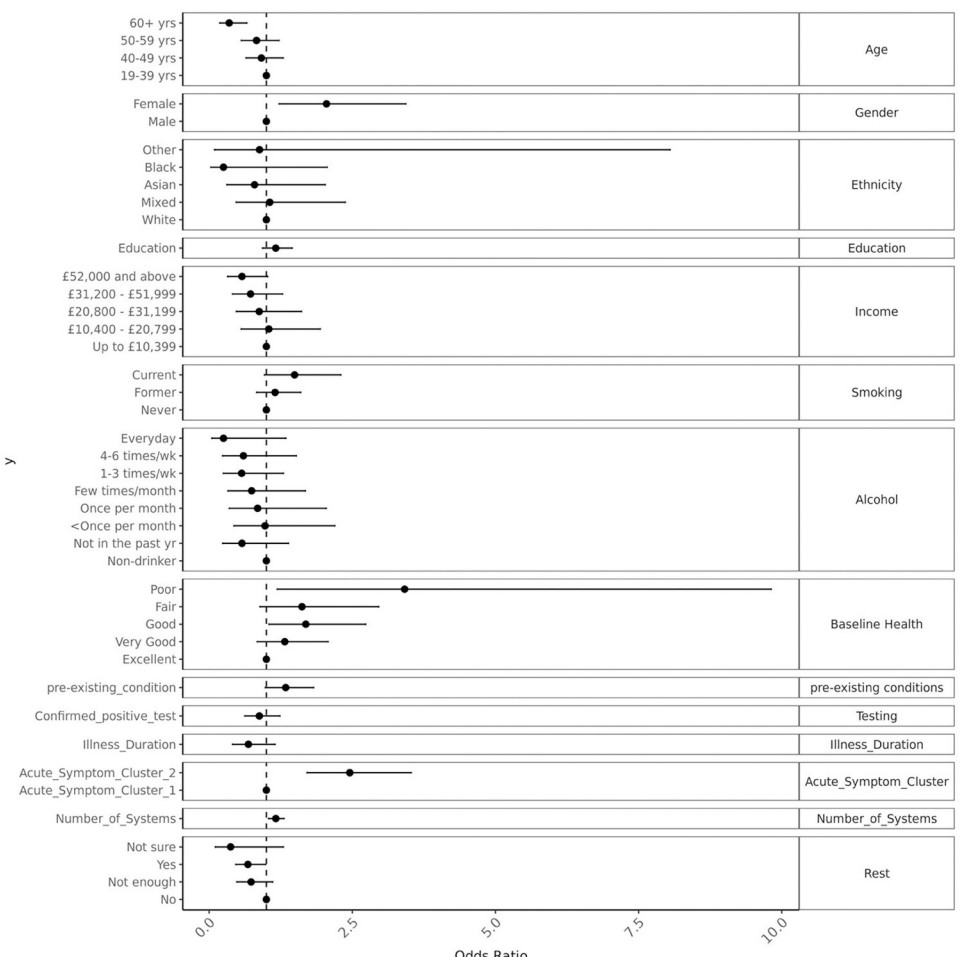

**Fig 4. Adjusted associations with developing multisystem ongoing symptom cluster (OSC) 2.**

system involvement, with those with more multisystem related symptoms more likely to move into the more severe cluster (S5 Fig).

Multivariate analysis suggested that gender and age were both predictors of transition from ASC1 to the more severe OSC2, with women being at higher risk (OR = 1.8; 95% CI 1.1, 3.2), and participants aged >60 years at lower risk (OR = 0.30, 95% CI 0.14–0.65). Number of systems with at least one associated symptom was also associated with higher likelihood of movement from ASC1 to OSC2 (OR = 1.1, 95% CI 1.0, 1.3), while having a confirmed positive test (OR = 0.66, 95% CI 0.44, 0.99) and having rested well during the first two weeks of the illness were associated with lower likelihood of movement from ASC1 to OSC2 (OR = 0.66, 95% CI 0.44, 0.99) (S6 Fig). Multivariate analysis was adjusted for duration of illness which was not associated with transition from ASC1 to OSC2. Month of infection was not included as a fixed effect in the analysis due to multi-collinearity with duration of illness.

## Discussion

Findings from this survey indicate that Long Covid is a debilitating multisystem illness for many of those experiencing it. Despite 9 in 10 of participants reporting good, very good, or excellent health before infection, a third said they were unable to live alone without assistance

at six weeks from onset. At an average of 7 months into Long Covid, 50% of participants said their illness affected self-care, 64% said it affected their mental health, and 75% said it affected their work. The majority of participants reported a fluctuating or relapsing/remitting pattern of illness. Two-thirds had to take time off sick from work with over a third reporting loss of income due to their illness. The symptoms of exhaustion, cognitive dysfunction, shortness of breath, headache, chest pressure/tightness, and muscle aches predominated. 86% of participants had a score of 4 or above on the Fatigue Severity Scale. For most participants, several of their initial symptoms became less prevalent with time, with the stark exception of cognitive dysfunction and palpitations. However, for a minority of participants who had extensive multisystem involvement from the start, many symptoms tended to become more common with time.

## Limitations

This is a non-representative survey which recruited through online support groups as well as generally through social media. The survey sampling method was convenience non-probability sampling. This means that the sample was not randomly drawn from the population of interest to ensure representativeness, and therefore the findings cannot be generalised to the groups not represented among participants, nor can they be used in any way to calculate the prevalence of Long Covid. Respondents were predominantly White, female and of higher socioeconomic status. People living with Long Covid who use social media (and therefore were able to access the survey) could have different characteristics to those who do not use such platforms. Indeed, some of those with Long Covid in the community who are suffering ill health may not realise it is due to Long Covid, particularly if their infection was not lab-confirmed in the first place. Data from the Office from National Statistics' Coronavirus Infection Survey, a large randomly-sampled survey of around 140,000 households from all parts of the UK, showed higher prevalence of Long Covid following confirmed or suspected infection in women, adults aged 35 to 69 years, and those living in most deprived areas, with no stark differences between ethnic groups [29, 30].

We tried to keep the survey as short as possible to be manageable, therefore some of the details around baseline characteristics, such as body mass index which requires self-measurement, were not collected. Although we asked about previous health status in general, we did not ascertain the prevalence/absence of each reported symptom before COVID-19. Given the variable severity and disability levels among participants at the later stages of the illness, there is also the possibility of recall bias in this survey, as the data about the acute stage and functional status at 6 weeks was collected retrospectively. The survey also aimed to collect data on those who have recovered from Long Covid and from short acute COVID-19 to allow comparisons of the acute symptoms between the two groups. However, the number of responses from these groups were too small to allow adequately powered comparisons. This likely reflects the motivation of individuals with Long Covid to participate in research which has Long Covid as its primary focus. Individuals with more symptoms or more severe symptoms may be more likely to respond to the survey.

Just over a quarter of survey participants reported having evidence of lab confirmation of COVID-19. However, the only pronounced differences in ongoing symptoms between those with lab confirmation and those without were the symptoms of loss/alteration of smell/taste. This is consistent with the other patient-led survey which included both confirmed and suspected cases of COVID-19 [10]. This difference can potentially be explained by people who have these symptoms being more likely to seek testing due to being specific to COVID-19 and heavily advertised in public health campaigns, as in the UK, unlike many of the other common

symptoms. Before loss of small/taste were added to the symptom lists, it could be that people experiencing them were more likely to seek healthcare input and hence get a test early on in their illness. It is important to note that people who reported testing negative were more likely to be tested much later in the illness than those who tested positive, again a consistent finding with Davis et al [10]. Also, the limitations of test accuracy and the importance of timing of testing (whether PCR or antibody) in relation to ascertaining SARS-CoV-2 infection are now well known [31, 32]. The sensitivity of PCR testing declines with time from the onset of infection, decreasing from 77% at 4 days to 50% by 10 days [33]. There is also emerging evidence that Long Covid is in itself linked to weak antibody response to SARS-CoV-2 and this could be implicated in the immunological process underlying this disease [21].

## Patient involvement

A major strength of this survey is that it was co-produced with people with Long Covid (pwLC). The idea for the survey came from pwLC and they were involved in the research from the initial discussions to the writing of the manuscript. It was important for us to ask the questions that reflect the main areas of concerns expressed by pwLC. NAA experienced Long Covid Symptoms and has been a strong advocate of the recognition and measurement of the condition [34, 35]. MEO and CH, as well as experiencing Long Covid themselves, have a wide overview of the symptoms, disability, disease course and concerns as expressed in the support groups and other national forums given their extensive involvement in Long Covid advocacy. The survey questions also received wider input from the members of the COVID-19 Facebook Research Involvement Group and went through several rounds of reshaping based on all the feedback.

Although we list including suspected as well as lab-confirmed cases in the survey as a limitation, we also consider this a strength of the survey. Community testing in the UK was stopped in early March 2020 but became available to essential workers in May 2020, and community testing was restarted in late Spring/Summer 2020 [25]. It is vital that people who got infected in the first wave of the pandemic and unable to access testing during the acute phase of their illness are included in research. They represent a big proportion of people currently living with Long Covid, and have the longest duration of illness, making it essential for studies about disease progression and prognosis to include them. We argue that those experiencing acute COVID-19 symptoms at a time of high national prevalence of infection, and not recovered for months after their acute episode, are very likely to have been infected with SARS-CoV-2 even if their access to lab testing was delayed or not possible at the time.

## Main findings and comparison to other data

Only 10% of survey participants reported less than good health prior to infection. The relapsing (comes and goes) or fluctuating nature of the illness was a prominent feature in most participants, but almost three quarters had daily symptoms. Many participants identified triggers for their symptoms, including physical or cognitive activity, stress, sleep disturbance and domestic chores. Avoiding the activities that trigger the symptoms mean adapting life routines accordingly. Some people may have life circumstances and job types that allow them to do that while others may not, leading to them feeling more unwell. This in turn has the potential to widen health and socioeconomic inequalities. Symptoms that were prevalent in the acute phase of the illness that were also common at the time of survey completion included exhaustion, breathlessness, headache, and chest pressure (heaviness) and/or tightness. Anxiety was reported by 28% and depression by 18% of participants. There are multiple reasons for anxiety in Long Covid including the unknown nature and prognosis of the illness, not having a

definitive treatment, and the anxiety of not being believed by others including health professionals and employers [36].

On clustering the ongoing symptoms, a minority cluster (OSC2: 11%) was detected with more multisystem involvement than the majority of participants (OSC1). In adjusted analysis, reporting sufficient rest during the first two weeks of the illness was associated with less likelihood of belonging to this cluster. It was also associated with less likelihood of moving from ASC1 to OSC2. Rest following acute infection is being recommended to prevent Long Covid [37]. However, taking weeks to recuperate is not always a choice for people who have pressing work or caring responsibilities, or those who are unable to take adequate sick leave because of limited employment rights or financial difficulties.

We stratified all of our main finding by test positivity. Most characteristics were similar between those who had lab confirmation and those who did not. A higher proportion of respondents who had a positive test reported that they rested well in the first two weeks of infection (60% vs 52%). This could be due to them recognising the seriousness of a COVID-19 illness having had a positive SARS-CoV-2 test and dedicating more time and resources towards their recovery and recuperation. This may in turn be linked to the finding that those who were lab-confirmed reported less functional disability, and a lower proportion of them reported loss of income (33% vs 39%). However, a considerable proportion of them were still severely functionally affected with 28% unable to live alone without help, 59% unable to perform usual activities and duties, 87% avoiding certain activities or duties at 6 weeks, and 28% with severe functional limitations. Additionally, the duration of loss of income due to illness experienced was higher among those with no test confirmation. As the majority of respondents reported still being ill at the time of completing the survey, this is likely to increase. This may mean that those who had lab-confirmation had an advantage in terms of both clinical recognition and for employment rights.

Descriptive findings from this survey on Long Covid are in line with findings from another online Long Covid survey which included participants from 56 countries with a majority from the United States [10]. Both surveys found that Long Covid symptoms affect multiple organ systems, with fatigue and cognitive dysfunction identified as the most common persistent symptoms. However, Davis et al collected data on more symptoms and thus identified a higher number of organ system involvement. Common triggers for return or exacerbation of symptoms were physical activity, cognitive activity, and stress, though our survey also identified sleep disturbance as a common trigger.

A study in Denmark following up 198 non-hospitalised PCR positive COVID-19 patients at 4 weeks and 129 at 12 weeks found similar findings to ours with fatigue and cognitive symptoms being the most common. There were no major differences in the prevalence of symptoms at these two time points other than loss of smell/taste being less common at 12 than 4 weeks from onset. Women and people with higher body mass index were more likely to suffer from persistent illness [16]. A study in the Faroe Islands of 180 mainly non-hospitalised PCR positive patients found that 20% had three or more symptoms after an average follow up of around 4 months, with the most prevalent symptoms being fatigue, loss of smell and taste and joint pains. In this study, they had a much higher proportion of participants with ongoing symptoms compared to acute for most of the symptoms, including fatigue [5]. Only 14% of our participants reported exhaustion as a new symptom not observed in the first two weeks of the illness, while 36% reported brain fog, 31% memory problems, and 27% poor concentration as symptoms they have not experienced in the first two weeks of the illness. It is possible that these symptoms were experienced in the first two weeks but because of the many other symptoms including fever, and people potentially being too ill to conduct cognitive tasks that require concentration, these were not specifically identified or recalled.

A study which recruited confirmed and suspected COVID-19 cases from Facebook groups in the Netherlands and Belgium found the average number of symptoms among non-hospitalised patients was 14, compared to an average of 12 initial and 10 ongoing symptoms in our survey. The most prevalent were similar to what we found including fatigue, shortness of breath, headache, and chest tightness, however cognitive dysfunction symptoms were not ascertained as an item in their questionnaire. These symptoms were included as open-text though not analysed [38]. In another paper from this study, it was reported that 52% of patients needed help with personal care more than two months from onset of symptoms, compared to before infection (8%) [39]. In our survey, 32% reported not being able to live alone without assistance at six weeks from onset of illness. At the time of completing the survey, a similar proportion (50%) said being ill affected their ability to self-care.

## Implications for research and practice

Many questions remain unanswered and require further research. Particular issues building on the findings from this survey include further understanding disease progression and studying the longitudinal clustering of symptoms and organ pathology. This is important to inform prognosis and prediction of progression at an early stage of the illness, which will in turn inform intensity and timing of appropriate interventions. The question of what pharmacological and non-pharmacological treatments work to 'cure' Long Covid or to improve quality of life and prevent complications also requires urgent research. The impact of Long Covid on disadvantaged socioeconomic and ethnic minority groups needs to be quantified. Potential mechanisms explaining why certain age or demographic groups may be more at risk need to be explored. Equitable, inclusive, and effective healthcare access is a fundamental right for all people living with Long Covid and must be systematically modelled to ensure services do not contribute to widening health disparities.

Long Covid studies based on both surveys and clinical records are needed as they complement each other. There is an assumption that Long Covid studies based on recruitment from primary care, Long Covid clinics, or clinical records data are unbiased compared to community surveys. However, although this assumption may be justified for other more established medical conditions, it does not necessarily apply to Long Covid [36]. Currently, healthcare access for Long Covid depends on many factors that may render healthcare research selective and unrepresentative. These include whether the person was tested or not, hospitalised or not, and their awareness that their own ill health may be linked to SARS-CoV-2 infection. This in turn, among other sociodemographic factors, will influence their health seeking behaviour. Also, clinicians' own variation in diagnosis and cognitive biases in the absence of objective guidelines on case definitions can play a part in who gets a diagnosis and gets coded in the medical records as Long Covid. Therefore, future applied research needs to triangulate the findings from representative community-based surveys, healthcare studies and qualitative research of patients' lived experiences.

The prevalence of Long Covid remains uncertain and dependent on the case definitions used and the duration of follow up. However, we know at this stage that it is not uncommon, including those whose infection was considered 'mild'. The number of cases will continue to increase if the virus continues to spread, therefore the issue of Long Covid and the impact it causes in terms of illness and disability is vital to pandemic and public health policy. This research demonstrates the impact of this prolonged illness on daily activities, work, physical, and mental health in a sample of predominantly healthy working-age individuals prior to infection. We explore how the acute symptoms are linked to the ongoing symptoms as a first step to help us characterise subgroups within the Long Covid umbrella. Long Covid is clearly a

multisystem disease, and individuals experiencing it must be able to receive care from a co-ordinated multidisciplinary team. The current model of Long Covid clinics in the UK will only be successful if there are clear, inclusive, and equitable referral pathways and case definitions [24, 40], and if effective and appropriately-resourced clinical input, investigations, treatments, and evidence-based rehabilitation become available.

## Supporting information

**S1 Fig. Reported SARS-CoV-2 testing history in survey participants.**
(DOCX)

**S2 Fig. Silhouette coefficient for 2 to 10 clusters.**
(DOCX)

**S3 Fig. Two clusters of acute symptoms and ongoing symptoms among these clusters.**
(DOCX)

**S4 Fig. Clustering of confirmed positive data only identifies similar clusters to whole dataset.**
(DOCX)

**S5 Fig. Transition from acute symptom cluster to ongoing symptom clusters by number of systems affected by symptoms.**
(DOCX)

**S6 Fig. Mutually adjusted predictors of transition from acute symptom cluster 1 (ASC1: Cardiopulmonary predominant) to ongoing symptom cluster 2 (OSC2: Multisystem).**
(DOCX)

**S1 Table. Classification of ongoing symptoms by organ system.**
(DOCX)

**S2 Table. Pre-existing conditions in survey participants.**
(DOCX)

**S3 Table. Symptoms categorised by phase of illness.**
(DOCX)

**S4 Table. Duration and pattern of illness in those who reported full recovery from Long Covid.**
(DOCX)

**S5 Table. Demographics and baseline health of survey participants.**
(DOCX)

**S6 Table. Initial symptoms experienced at the start of COVID-19 illness (first two weeks).**
(DOCX)

**S7 Table. Ongoing symptoms, fatigue severity and organ systems affected.**
(DOCX)

## Acknowledgments

We thank all participants for their time and commitment completing this survey. We also sincerely thank members of the COVID-19 Research Involvement Group for providing feedback

on earlier versions of the questionnaire. Margaret E O'Hara, Claire Hastie and Nisreen A Alwan experience(d) Long Covid symptoms.

## Author Contributions

**Conceptualization:** Nida Ziauddeen, Margaret E. O'Hara, Claire Hastie, Guiqing Yao, Nisreen A. Alwan.

**Data curation:** Nida Ziauddeen, Nisreen A. Alwan.

**Formal analysis:** Nida Ziauddeen, Deepti Gurdasani.

**Investigation:** Nida Ziauddeen, Nisreen A. Alwan.

**Methodology:** Nida Ziauddeen, Deepti Gurdasani, Margaret E. O'Hara, Claire Hastie, Paul Roderick, Guiqing Yao, Nisreen A. Alwan.

**Writing – original draft:** Nida Ziauddeen, Deepti Gurdasani, Nisreen A. Alwan.

**Writing – review & editing:** Nida Ziauddeen, Deepti Gurdasani, Margaret E. O'Hara, Claire Hastie, Paul Roderick, Guiqing Yao, Nisreen A. Alwan.

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
