## [Decision Letter · Decision Letter 0]

2 Nov 2021

PONE-D-21-14682Characteristics of Long Covid: findings from a social media surveyPLOS ONE

Dear Dr. Ziauddeen,

Thank you for submitting your manuscript to PLOS ONE. After careful consideration, we feel that it has merit but does not fully meet PLOS ONE’s publication criteria as it currently stands. Therefore, we invite you to submit a revised version of the manuscript that addresses the points raised during the review process.

ACADEMIC EDITOR:

The reviewers raise important issues. While Reviewer #1 had concerns about the data that the authors cannot address and already acknowledge in the discussion, please do the following to address comments #1-3:

- expand the discussion of the non-representativeness of the sample to contrast it with the demographic characteristics of cases in the UK at the time

- acknowledge that those with more symptoms and more severe symptoms may be more likely to respond

- the lack of lab confirmation is discussed and justified but is still concerning. Is there any data on what else was circulating at the time (e.g. influenza) in the UK that could be added to the discussion to feel more confident that these were COVID cases? It would also be helpful to see separate out test negatives and not tested for Tables 1-3 to see if the not tested were more similar to the test positives or negatives - these can be added as supplemental tables.

Please address or respond to all other comments from reviewer #2 and #3.

Additional minor comments to be addressed:

- healthcare utilization - please ensure to use language that is not specific to the UK or provide an explanation (e.g. 111 calls - are these emergency calls?)

- Figure 2 - I assumed that the x-axis groups (reasons) were the strata but in fact the series groups (work pattern) are the strata and add up to 100% - consider switching

- add sample sizes to supplementary figure 1 so that it can be matched up with the text in the results

- unclear how Supplementary Figure 3 and Figure 3 are different

- Review titles for each panel in figure 3 as they all include 'ongoing' and are hard to match up with the groups. Add the acronyms used in the text (ASC1/ASC2/OSC1/OSC2).

- Discussion - the authors mention recall bias as it relates to the acute stage, but this also applies to functional status at 6 weeks as most participants are far beyond 6 weeks of illness and should be mentioned.

We look forward to receiving your revised manuscript.

Kind regards,

Catherine G. Sutcliffe

Academic Editor

PLOS ONE

Journal Requirements:

Reviewers' comments:

Reviewer's Responses to Questions

**Comments to the Author**

1. Is the manuscript technically sound, and do the data support the conclusions?

Reviewer #1: Partly

Reviewer #2: Partly

Reviewer #3: Yes

2. Has the statistical analysis been performed appropriately and rigorously? 

Reviewer #1: N/A

Reviewer #2: I Don't Know

Reviewer #3: No

3. Have the authors made all data underlying the findings in their manuscript fully available?

Reviewer #1: Yes

Reviewer #2: Yes

Reviewer #3: Yes

4. Is the manuscript presented in an intelligible fashion and written in standard English?

Reviewer #1: Yes

Reviewer #2: Yes

Reviewer #3: No

5. Review Comments to the Author

Reviewer #1: Summary: This manuscript summarizes the findings from a social media survey that was done in November 2020 to assess the symptoms of persons with Long COVID and to identify symptom clusters among those with long COVID.

Study Strengths: The study is well written and done a by a team that includes persons with Long COVID, researchers, and researchers with Long COVID. It appropriately starts the discussion by addressing the limitations of the study design and the response among those who are white, of higher SES and were able to use social media The authors have identified several clusters of symptoms and tried to estimate the impact on missed work/life experiences by those who are affected with long COVID in the survey respondents.

Major comments: I have several major concerns with this paper, including the demographics of the sample, potential for recall bias, and the lack of laboratory evidence of COVID infection in 75% of the respondents.

1. This study was done through social media and the respondents are for the most part, highly educated, white women which severely limits its generalizability given the disproportionate impact of COVID-19 in persons of color and the relatively equal distribution of COVID by sex.

2. I also worry about response bias, as only 2.3% of participants felt that they had recovered to baseline health, which far exceeds the approximately 30% of those with COVID who develop long term sequelae. It appears as though those with more severe symptoms were more likely to respond greatly skewing the estimates of this condition. The authors appropriately note that next steps to evaluate the prevalence, predictors and prognosis will need a more representative population and a standardized case definition.

3. In addition, of the 2550 participants in the study, only 26% had established laboratory evidence of COVID-19. It is also striking that loss of sense of taste/smell were not among the most common symptoms reported in those without lab confirmed infection, yet over 25% of those with lab confirmed infection has loss of sense of taste/or smell. While it is possible that many of those without laboratory testing did have COVID, the fact that the majority reported nonspecific symptoms of fatigue, headache, myalgias and chest tightness, but did not have any impact on taste or smell concerns me that we are assessing another illness or comorbidity which may have had an onset at the same time as the pandemic onset. This is also consistent with the time off from work, where those who had laboratory confirmation of their illness had less time off than those without laboratory confirmation (129 vs. 84 days) and the need for assistance because of the inability to live alone.

4. Given the changes in laboratory testing with time, were there any changes in the positivity rate of those with symptoms or the correlation of PCR positivity with Ab positivity with month of illness (ie those who tested in late summer were more or less likely to have concordant results rather than those who tested early on when testing was harder to get and less reliable?) Why are 29% of those who had a PCR negative for antibodies? (Confirming these antibodies were checked pre-vaccine?)

Minor comments:

The authors noted that 12% were admitted to the hospital after 2 weeks from onset of illness. What were the reasons for these hospitalizations and what was the median time from onset of COVID symptoms to hospitalization?

I am concerned that the multivariate analysis is of limited utility given the high proportion of women completing the study vs. men. It also appears that there is an association (albeit non-significant) between baseline health status and development of OSC2. This is not surprising, and should be further investigated.

Reviewer #2: General:

Although this survey appears to be nearly a year old, the overall topic remains quite timely and of great relevance. This article is a nice addition to the LongCOVID literature in that it includes a rich dataset and wide array of analyses. Further clarifying the approach in the abstract, more clearly presenting the comparison between lab-confirmed and not in the results, and better describing the trends in symptoms over time would strengthen this article.

Abstract:

The main manuscript does a great job laying out the approach, methods, results, and limitations and is also much clearer about the engagement of the pwLC in the process and the recruitment methods. The abstract is much harder to follow. It would help to be more overt that the study specifically targeted populations who already identified themselves as having Long COVID. The methods should include the basic statistical analysis) (descriptive, comparison between lab-confirmed and not) in addition to the approach to clustering as the former makes up a good portion of the paper. The flow of the results could be improved. It would help to know much earlier that only 26.5% of the participants were lab confirmed (would more closely mirror the flow in the manuscript). Rather than use the term “Biggest difference” it would be good to clarify if this is the only statistically significant difference in reported symptoms between these two groups and what was the rate that this was reported in each group. Please clarify the difference between fluctuating and relapsing in the manuscript.

Introduction:

Potentially of interest, just this week the WHO released a case definition of post COVID conditions – you may want to consider citing this

Pg 4- It sounds like the initial symptoms are meant to be initial symptoms of COVID-19 and then ongoing symptoms are ongoing symptoms of Long COVID. Please confirm. Please also more explicitly define “ongoing” somewhere. Is that >4 weeks or ongoing at the time of the survey?

Methods:

A fair amount of language in this section seems more appropriate for the discussion section.

Pg 5 – It would be good to more clearly use the term “Self-reported” for symptoms and diagnosis, both here and in the abstract

Results:

In the methods, you noted that comparison was made between lab-confirmed and not – if there is space, it would help to have that comparison included in each section. (Could consider reducing the length of the discussion section some to allow for this)

You also note a large number of persons who tested negative, specifically comparing the positive and negative groups and assessing for differences there would also be of interest (though acknowledging that is a lot more work, I would see that as optional and just encourage the team to look at this)

Pg 7 - Please clarify is the median duration of illness, was median time since illness, or median duration of symptoms

Please clarify if “ongoing’ symptoms in the methods and results indicates at >4 wks or at the time of the survey. It would be helpful to include an overall overview of how symptoms decreased over time if they did this, such as xx (%) persons reported one or more ongoing symptoms for > 4weeks, xx (%) for > 12-weeks, and xx (%) for > 6 months following their initial infection.

Pg 11 – for two clusters, it might help to identify the most commonly reported symptoms in each cluster in the manuscript

Discussion:

The summary of evolution of symptoms over time at the end of the first paragraph was helpful and not as clear in the results section. In general this section is quite verbose and could likely be written more concisely in order to allow for some of the recommendations above.

Tables and Figures:

Table 1 - Would recommend splitting "Tested negative" and "not tested" if possible

Table 3 - There are a number of symptoms listed here, for which there appears to be a statistically significant difference in frequency between those who tested positive and those who did not, that weren't highlighted in the manuscript. Would recommend more explicitly including/listing these as well.

Reviewer #3: The authors analyze data generated from a cross-sectional, online survey exploring long Covid features in subjects from the UK. The paper has significant amount of material, and the study is timely, and relevant. My comments are as follows:

(a) The purpose of the Abstract will be better served if re-written as Background, Methods, Results, and Conclusions. Otherwise, the long writeup doesn't appear very pleasing to go through.

(b) Statistical analysis: t-tests were used for continuous variables. How was the assumption of Normality checked? Under violations, better to resort to 2-sample Wilcoxon tests.

(c) Statistical analysis: Hierarchical agglomerative clustering (HAC) was used, utilizing the complete method. More details are needed on what that is. A variety of other methods, such as Ward, single-linkage, etc are available. Why were they not used? Justify.

(d) Statistical analysis: HAC do not work for missing data, and can be quite sensitive to the choice of the distance/dissimilarity matrix employed. A sensitivity analysis, however small, would be very relevant here. If not, justification is needed behind the choice of the specific dissimilarity matrix used.

(e) Statistical analysis: "Multivariable" logistic regression was used. The correct word is "Multiple" logistic regression, because multivariable would mean something different. This change needs to be made throughout the manuscript. Furthermore, some goodness-of-fit assessments after performing the multiple logistic regressions (say, via the Hosmer-Lemeshow statistics) is desirable.

6. PLOS authors have the option to publish the peer review history of their article (what does this mean?). If published, this will include your full peer review and any attached files.

Reviewer #1: No

Reviewer #2: No

Reviewer #3: No

---

## [Author Response · Author response to Decision Letter 0]

23 Dec 2021

Thank you for the helpful feedback which we address point by point below. Since it has been a long while since we submitted this manuscript to PLOS One, we have also gone through the whole paper and updated information and references when necessary. We have changed some of the preprints referenced to their current journal citations if relevant. The original referencing and text remains in the preprint version of our paper:

https://www.medrxiv.org/content/10.1101/2021.03.21.21253968v2.full.pdf

The reviewers raise important issues. While Reviewer #1 had concerns about the data that the authors cannot address and already acknowledge in the discussion, please do the following to address comments #1-3:

- expand the discussion of the non-representativeness of the sample to contrast it with the demographic characteristics of cases in the UK at the time

Thank you, we have expanded the discussion comparing to an earlier release from the UK’s Office for National Statistics on the prevalence of Long Covid (lines 475-479 in the clean and 494-498 in the tracked version):

“Data from the Office from National Statistics’ Coronavirus Infection Survey, a large randomly-sampled survey of around 140,000 households from all parts of the UK, showed higher prevalence of Long Covid following confirmed or suspected infection in women, adults aged 35 to 69 years, and those living in most deprived areas, with no stark differences between ethnic groups28,29”.

- acknowledge that those with more symptoms and more severe symptoms may be more likely to respond

Thank you, we have now added this sentence in the discussion (lines 490-491 in the clean and 516-517 in the tracked version). 

- the lack of lab confirmation is discussed and justified but is still concerning. Is there any data on what else was circulating at the time (e.g. influenza) in the UK that could be added to the discussion to feel more confident that these were COVID cases? It would also be helpful to see separate out test negatives and not tested for Tables 1-3 to see if the not tested were more similar to the test positives or negatives - these can be added as supplemental tables.

Thank you for the suggestion, we have now included the test negative/not tested split in the supplementary tables 5-7. We believe there are three factors that are essential to consider here: the timing and availability of covid testing, and the potential for the pathological processes involved in Long Covid to influence the direction of the test result. 

As we state in the results section: “The date of PCR test was available for 1491 of these 1582 participants. Twenty percent (n=304) were first tested at 0-5 days of onset of symptoms, with 72.7% of this group testing positive, while 80% (n=1187) were first tested ≥6 days of onset of symptoms, with 12.6% of this group testing positive.” We know that the sensitivity of PCR declines with time from onset of infection, decreasing from 77% at 4 days to 50% by 10 days (Hellewell J, Russell TW, Matthews R, Severn A, Adam S, Enfield L, et al. Estimating the effectiveness of routine asymptomatic PCR testing at different frequencies for the detection of SARS-CoV-2 infections. BMC Medicine. 2021 Apr 27;19(1):106). 

With regards to the availability of testing, 73% of respondents reported first developing symptoms before June 2020 in the UK when community testing was not available for those not admitted to hospital or not in health/social care. We touch on this in the discussion under the ‘Patient involvement’ subheading, and actually consider the inclusion of those with suspected infection a strength of the study: “Although we list including suspected as well as lab-confirmed cases in the survey as a limitation, we also consider this a strength of the survey. Community testing in the UK was stopped in early March 2020 but became available to essential workers in May 2020, and community testing was restarted in late Spring/Summer 202024. It is vital that people who got infected in the first wave of the pandemic and unable to access testing during the acute phase of their illness are included in research. They represent a big proportion of people currently living with Long Covid, and have the longest duration of illness, making it essential for studies about disease progression and prognosis to include them. We argue that those experiencing acute COVID-19 symptoms at a time of high national prevalence of infection, and not recovered for months after their acute episode, are very likely to have been infected with SARS-CoV-2 even if their access to lab testing was delayed or not possible at the time.” The senior author has written extensively, solely and with others, on the potential serious inequalities in recognition and support resulting from that unavailability of testing at the start of the pandemic and from only including those who have evidence of a positive test in Long Covid care, surveillance, and research, for example:

Alwan NA. A negative COVID-19 test does not mean recovery. Nature. 2020 Aug 11;584(7820):170–170.

Alwan NA. The road to addressing Long Covid. Science. 2021 Jul 30;373(6554):491–3.

Alwan NA, Attree E, Blair JM, Bogaert D, Bowen M-A, Boyle J, et al. From doctors as patients: a manifesto for tackling persisting symptoms of covid-19. BMJ. 2020 Sep 15;370:m3565.

Alwan NA. The teachings of Long COVID. Commun Med. 2021 Jul 12;1(1):1–3.

Alwan NA, Johnson L. Defining long COVID: Going back to the start. Med. 2021 Mar 25;2(5):501–4.

We consider it very important for research published in medical journals, which plays a core role in shaping clinical pathways and agendas, to factor in this inequality angle. 

The third point is specific to antibody testing. There is emerging evidence that Long Covid is in itself linked to weak antibody response to SARS-CoV-2 and this could be implicated in the immunological process underlying this disease. We have cited a reference to support this claim (lines 505-507 in the clean and 531-533 in the tracked version): García-Abellán J, Padilla S, Fernández-González M, García JA, Agulló V, Andreo M, et al. Antibody Response to SARS-CoV-2 is Associated with Long-term Clinical Outcome in Patients with COVID-19: a Longitudinal Study. J Clin Immunol. 2021 Oct 1;41(7):1490–501.

Please address or respond to all other comments from reviewer #2 and #3.

Additional minor comments to be addressed:

- healthcare utilization - please ensure to use language that is not specific to the UK or provide an explanation (e.g. 111 calls - are these emergency calls?)

111 is a non-emergency medical care number that guides callers to the right service. The number is specific to the UK. We have deleted reference to 111 calls and updated this to “calls to non-emergency medical care number” (lines 346-347 in the clean and 364-365 in the tracked version).

- Figure 2 - I assumed that the x-axis groups (reasons) were the strata but in fact the series groups (work pattern) are the strata and add up to 100% - consider switching

Thank you, we have switched the strata as suggested. 

- add sample sizes to supplementary figure 1 so that it can be matched up with the text in the results

Thank you, we have added sample sizes to supplementary figure 1.

- unclear how Supplementary Figure 3 and Figure 3 are different

Figure 3 presents the acute (bottom panels) and ongoing (top panels) symptoms by the two ongoing symptom clusters. Supplementary Figure 3 presents the same by the two acute symptom clusters.

- Review titles for each panel in figure 3 as they all include 'ongoing' and are hard to match up with the groups. Add the acronyms used in the text (ASC1/ASC2/OSC1/OSC2).

Thank you, we have the appropriate acronyms used in the text (OSC1/OSC2). The top panels present ongoing symptoms in the ongoing clusters and the bottom two panels present acute symptoms in the ongoing clusters.

- Discussion - the authors mention recall bias as it relates to the acute stage, but this also applies to functional status at 6 weeks as most participants are far beyond 6 weeks of illness and should be mentioned.

Thank you, we have added this to the sentence on recall bias (line 485 in the clean and 511 in the tracked version).

Reviewers' comments:

Reviewer #2: 

General: Although this survey appears to be nearly a year old, the overall topic remains quite timely and of great relevance. This article is a nice addition to the Long COVID literature in that it includes a rich dataset and wide array of analyses. Further clarifying the approach in the abstract, more clearly presenting the comparison between lab-confirmed and not in the results, and better describing the trends in symptoms over time would strengthen this article.

Abstract:

The main manuscript does a great job laying out the approach, methods, results, and limitations and is also much clearer about the engagement of the pwLC in the process and the recruitment methods. The abstract is much harder to follow. It would help to be more overt that the study specifically targeted populations who already identified themselves as having Long COVID. The methods should include the basic statistical analysis) (descriptive, comparison between lab-confirmed and not) in addition to the approach to clustering as the former makes up a good portion of the paper. The flow of the results could be improved. It would help to know much earlier that only 26.5% of the participants were lab confirmed (would more closely mirror the flow in the manuscript). Rather than use the term “Biggest difference” it would be good to clarify if this is the only statistically significant difference in reported symptoms between these two groups and what was the rate that this was reported in each group. Please clarify the difference between fluctuating and relapsing in the manuscript.

Thank you, we have updated the abstract as suggested. We have added subheadings, amended the methods as suggested above, and moved the percentage lab confirmed up as suggested.

There are several statistically significant differences between the two groups. We highlighted loss of sense of smell/taste as the reported prevalence as an ongoing symptom in those that tested positive (24.4% smell, 20.9% taste) is over double that in those that tested negative/not tested (10.2% smell, 9.2% taste) which is statistically significant but also a big difference between the groups. For other statistically significant differences between symptoms, the difference between the groups was usually around 10% or less. These are listed in table 3 and include: sore throat, abdominal pain, nausea, brain fog, chest pain, chest tightness, chills, hoarse voice and a feeling of pins and needles. We have added these to the text of the results section (lines 258-262 in the clean and 282-286 in the tracked version), as well as differences between the groups in acute symptoms too (lines 266-270 in the clean and 272-277 in the tracked version) but have removed reference to differences from the abstract to fit in within the word count.

We have clarified the difference between fluctuating (intensity of symptoms changes but symptoms never completely go away) and relapsing (experience symptom-free periods in between relapses) in the results section of the manuscript (lines 293-295 in the clean and 310-312 in the tracked version).

Introduction:

Potentially of interest, just this week the WHO released a case definition of post COVID conditions – you may want to consider citing this

Thank you for highlighting this. The senior author was part of the WHO clinical case definition working group as cited in the report. The WHO definition document is now cited in the last paragraph of the discussion (line 673): A clinical case definition of post COVID-19 condition by a Delphi consensus, 6 October 2021 [Internet]. [cited 2021 Oct 15]. Available from: https://www.who.int/publications-detail-redirect/WHO-2019-nCoV-Post_COVID-19_condition-Clinical_case_definition-2021.1

Pg 4- It sounds like the initial symptoms are meant to be initial symptoms of COVID-19 and then ongoing symptoms are ongoing symptoms of Long COVID. Please confirm. Please also more explicitly define “ongoing” somewhere. Is that >4 weeks or ongoing at the time of the survey?

Thank you. Initial symptoms are symptoms experienced in the first two weeks of infection. Ongoing symptoms are symptoms that remained or developed after the acute stage (≥2 weeks) and continue to be experienced in the longer term over the course of the illness. We have clarified this in the methods section (lines 151-154 in the clean and 164-167 in the tracked version). 

Methods:

A fair amount of language in this section seems more appropriate for the discussion section.

Thank you. If the reviewer is referring to the description of how the survey was co-produced with patients, we feel this needs to stay in the methods section. 

Pg 5 – It would be good to more clearly use the term “Self-reported” for symptoms and diagnosis, both here and in the abstract

We have added the term ‘self-reported’ to the abstract (line 28 in the clean and 33-34 in the tracked version). Symptoms are by definition self-reported as there is no other way to measure them. 

Results:

In the methods, you noted that comparison was made between lab-confirmed and not – if there is space, it would help to have that comparison included in each section. (Could consider reducing the length of the discussion section some to allow for this)

Thank you, we have added comparison in each section as suggested (lines 253-255, 272-289, 306-309, 313-316, 323-234, 333-334, 342-344 and 355-356, all in the tracked version).

You also note a large number of persons who tested negative, specifically comparing the positive and negative groups and assessing for differences there would also be of interest (though acknowledging that is a lot more work, I would see that as optional and just encourage the team to look at this)

Table 1 - Would recommend splitting "Tested negative" and "not tested" if possible

Thank you for the suggestion, we have split the tested negative/not tested group and presented in the supplemental tables (5-7). We have chosen to retain the tables as is in the main analysis. Please see our response above to a similar comment by the editor/reviewer 1.

Pg 7 - Please clarify is the median duration of illness, was median time since illness, or median duration of symptoms

Median duration of illness is the duration of experiencing symptoms. We have added this in brackets to clarify (line 233 in the clean and 247 in the tracked version).

Please clarify if “ongoing’ symptoms in the methods and results indicates at >4 wks or at the time of the survey. It would be helpful to include an overall overview of how symptoms decreased over time if they did this, such as xx (%) persons reported one or more ongoing symptoms for > 4weeks, xx (%) for > 12-weeks, and xx (%) for > 6 months following their initial infection.

Ongoing symptoms are symptoms experienced in the longer-term over the course of the illness which could be anytime past the first 2 weeks (lines 153-154 in the clean and 174-175 in the tracked version). We did not collect data on the duration that each symptom was experienced for so cannot include the suggested overview. 

Pg 11 – for two clusters, it might help to identify the most commonly reported symptoms in each cluster in the manuscript

Thank you, we have added the most commonly reported symptoms in the manuscript (lines 383-389 and 395-402 in the clean, and 401-407 and 414-421 in the tracked version). 

Discussion:

The summary of evolution of symptoms over time at the end of the first paragraph was helpful and not as clear in the results section. In general this section is quite verbose and could likely be written more concisely in order to allow for some of the recommendations above.

Thank you, we have cut down this section in several places where we felt possible/appropriate. 

Table 3 - There are a number of symptoms listed here, for which there appears to be a statistically significant difference in frequency between those who tested positive and those who did not, that weren't highlighted in the manuscript. Would recommend more explicitly including/listing these as well.

Thank you, we have highlighted these differences in the manuscript. Please see response to editors/other reviewers on this above (lines 262-270 in the clean and 277-286 in the tracked version). 

Reviewer #3: 

The authors analyze data generated from a cross-sectional, online survey exploring long Covid features in subjects from the UK. The paper has significant amount of material, and the study is timely, and relevant. My comments are as follows:

(a) The purpose of the Abstract will be better served if re-written as Background, Methods, Results, and Conclusions. Otherwise, the long writeup doesn't appear very pleasing to go through.

Thank you, we have added the suggested sections to the abstract.

(b) Statistical analysis: t-tests were used for continuous variables. How was the assumption of Normality checked? Under violations, better to resort to 2-sample Wilcoxon tests.

Thank you, we used t-tests or Mann Whitney U test for continuous variables. We had added this as a footnote to the tables where relevant (Tables 2, 3 and 6) but had not updated the methods section. We have now added this to the methods (line 181 in the clean and 194 in the tracked version).

(c) Statistical analysis: Hierarchical agglomerative clustering (HAC) was used, utilizing the complete method. More details are needed on what that is. A variety of other methods, such as Ward, single-linkage, etc are available. Why were they not used? Justify.

We used multiple methods for clustering in our early analysis. We examined K-medoids clustering, and Gaussian mixture model clustering (to allow for non-spherical clusters). We assessed the optimal number of clusters using both these methods with the silhouette method (for K-medoids) and Bayesian information criteria (BIC) for Guassian mixture clustering, and the stability of clusters using Jaccard’s coefficient. While K-medoids clustering identified two clusters, the Jaccard’s coefficient was below 0.75 suggesting these clusters were unstable. With Gaussian mixture clustering, we evaluated BIC for up to 15 clusters. However, BIC continued to increase with the number of clusters, suggesting a very high number of clusters would be optimal. Given the instability of clusters, and the need for better interpretability of clusters, we chose to carry out exploratory clustering using hierarchical agglomerative clustering rather than apply methods with unstable clusters, or potentially very large numbers of clusters that would hamper interpretability. We consider these methods exploratory rather than an exhaustive evaluation of clustering of these data. Given different approaches to HAC can lead to different clusters, we used the complete method, average method, Ward’s method, and single method. Using these, we noted that the single method was not optimal as it led to single individuals being members of clusters. Among other methods, the correlation between all methods was moderate to high but this was highest between the complete method and the average method (r=0.7), so we chose the complete method for our exploratory analysis. We note that our clusters in addition to correlating strongly with functional indicators, also accord well with the clusters obtained from other studies such as REACT-1 providing evidence for their clinical significance, and consistency with other studies. 

(d) Statistical analysis: HAC do not work for missing data, and can be quite sensitive to the choice of the distance/dissimilarity matrix employed. A sensitivity analysis, however small, would be very relevant here. If not, justification is needed behind the choice of the specific dissimilarity matrix used.

We note that there was very little missing data in our survey. While we carried out a complete data analysis, we note that only 10 individuals (less than 0.4% of our sample was excluded due to missing data). We do not consider that exclusion of such a small number of individuals would have altered our results. We used the Gower’s dissimilarity matrix given the categorical nature of data, as this is appropriate for such data. 

(e) Statistical analysis: "Multivariable" logistic regression was used. The correct word is "Multiple" logistic regression, because multivariable would mean something different. This change needs to be made throughout the manuscript. Furthermore, some goodness-of-fit assessments after performing the multiple logistic regressions (say, via the Hosmer-Lemeshow statistics) is desirable.

Thank you, we have updated this as suggested but multivariable and multiple logistic regression are different words for the same (https://www.ncbi.nlm.nih.gov/pmc/articles/PMC3518362/) (line 212 in the clean and 226 in the tracked version).

---

## [Decision Letter · Decision Letter 1]

27 Jan 2022

PONE-D-21-14682R1Characteristics and impact of Long Covid: findings from an online surveyPLOS ONE

Dear Dr. Ziauddeen,

Thank you for submitting your manuscript to PLOS ONE. After careful consideration, we feel that it has merit but does not fully meet PLOS ONE’s publication criteria as it currently stands. Therefore, we invite you to submit a revised version of the manuscript that addresses the points raised during the review process.

ACADEMIC EDITOR: The authors have adequately addressed the reviewer's comments. Before the paper can be accepted, please address the following minor issues:

1. Abstract - Methods: Suggest revising to “We collected self-reported data through an online survey using convenience non-probability sampling. The survey enrolled adults with lab-confirmed (PCR or antibody) or suspected COVID-19 who were not hospitalized in the first two weeks of illness. This analysis was restricted to those with self-reported Long Covid. Univariate comparisons..."

2. Introduction: typo on line 108 of tracked version - “perceived a lack of data on COVID-19 sequelae…”

3. Introduction – last paragraph: suggest revising line 110 in tracked version to "In adults who self-reported Long Covid after suspected or confirmed COVID-19 and..."

4. Methods – last paragraph: suggest revising line 238 in tracked version to “As the full analysis included those with and without lab-confirmed diagnosis of COVID-19, we examined whether this was a significant predictor of cluster membership to assess whether clusters correlated with having lab-confirmation of infection. We also carried out an additional sensitivity analysis by clustering only those with lab confirmation to see if clusters obtained were different from the full sample analysis.”

5. Results, new sentence in line 256-258 of tracked version: Based on Table 1, I think you mean: “The proportion of participants from outside the UK was higher among those with lab-confirmed infection (29.1%) than among those with suspected infection (17.0%).”

6. Results – first paragraph: can the authors add in the median time from symptom onset to completing the survey (presumably this is different from reported duration of illness since some people have recovered). I am also assuming that duration of illness was just the time from symptom onset to completing the questionnaire for those still symptomatic at survey completion (if this is incorrect, then please clarify in the methods how this was calculated)- was the longer duration of illness among negatives simply due to a longer time since survey completion?  

7. Table 1 and 2 footnote: revise to “Comparisons between those with and without lab-confirmed COVID-19 used...”

8. Results – previous health: revise last sentence to “There were no significant differences in these proportions between those with and without lab-confirmed infection”.

9. Results, course of illness: revise line 287-288 of tracked version to “…which were higher in those with lab confirmation than without, whereas abdominal pain, nausea, chest pain, chest tightness, chills, hoarse voice, sore throat, sneezing and pins and needles were lower in those with lab-confirmation than without.”

10. Table 4: ‘Comes and goes’ and ‘relapsing’ are listed separately in the table but reported/defined together in the text (line 314). How are these different?

11. Results – lab confirmation of infection: Move sentence (starting with “1172 participants (46%) reported having an antibody test…”) on line 381 of tracked version to the beginning of that paragraph on antibody testing. Also add a period between ‘test’ and ’26.3%’ on line 379.

12. Table S4 – many of the categories are overlapping – please revise.

13. Results – lab confirmation of infection: the last paragraph seems redundant with the prior sections since differences by lab confirmation are now presented throughout – suggest deleting it.

14. Results, clustering: second paragraph describing results of univariate analysis – for some the % are presented in the text (e.g. for loss of income)- suggest removing to be consistent for all results mentioned.

15. Results, clustering: last paragraph, correct reference to “OR” for ‘having a confirmed positive test’.

We look forward to receiving your revised manuscript.

Kind regards,

Catherine G. Sutcliffe

Academic Editor

PLOS ONE

Reviewers' comments:

Reviewer's Responses to Questions

**Comments to the Author**

1. If the authors have adequately addressed your comments raised in a previous round of review and you feel that this manuscript is now acceptable for publication, you may indicate that here to bypass the “Comments to the Author” section, enter your conflict of interest statement in the “Confidential to Editor” section, and submit your "Accept" recommendation.

Reviewer #3: All comments have been addressed

2. Is the manuscript technically sound, and do the data support the conclusions?

Reviewer #3: (No Response)

3. Has the statistical analysis been performed appropriately and rigorously? 

Reviewer #3: (No Response)

4. Have the authors made all data underlying the findings in their manuscript fully available?

Reviewer #3: (No Response)

5. Is the manuscript presented in an intelligible fashion and written in standard English?

Reviewer #3: (No Response)

6. Review Comments to the Author

Reviewer #3: (No Response)

7. PLOS authors have the option to publish the peer review history of their article (what does this mean?). If published, this will include your full peer review and any attached files.

Reviewer #3: No

---

## [Author Response · Author response to Decision Letter 1]

1 Feb 2022

ACADEMIC EDITOR: The authors have adequately addressed the reviewer's comments. Before the paper can be accepted, please address the following minor issues:

1. Abstract - Methods: Suggest revising to “We collected self-reported data through an online survey using convenience non-probability sampling. The survey enrolled adults with lab-confirmed (PCR or antibody) or suspected COVID-19 who were not hospitalized in the first two weeks of illness. This analysis was restricted to those with self-reported Long Covid. Univariate comparisons..."

Thank you, we have revised as suggested.

2. Introduction: typo on line 108 of tracked version - “perceived a lack of data on COVID-19 sequelae…”

Thank you, we have corrected this typo.

3. Introduction – last paragraph: suggest revising line 110 in tracked version to "In adults who self-reported Long Covid after suspected or confirmed COVID-19 and..."

Thank you, we have revised as suggested.

4. Methods – last paragraph: suggest revising line 238 in tracked version to “As the full analysis included those with and without lab-confirmed diagnosis of COVID-19, we examined whether this was a significant predictor of cluster membership to assess whether clusters correlated with having lab-confirmation of infection. We also carried out an additional sensitivity analysis by clustering only those with lab confirmation to see if clusters obtained were different from the full sample analysis.”

Thank you, we have revised as suggested.

5. Results, new sentence in line 256-258 of tracked version: Based on Table 1, I think you mean: “The proportion of participants from outside the UK was higher among those with lab-confirmed infection (29.1%) than among those with suspected infection (17.0%).”

Thank you, we have revised as suggested.

6. Results – first paragraph: can the authors add in the median time from symptom onset to completing the survey (presumably this is different from reported duration of illness since some people have recovered). I am also assuming that duration of illness was just the time from symptom onset to completing the questionnaire for those still symptomatic at survey completion (if this is incorrect, then please clarify in the methods how this was calculated)- was the longer duration of illness among negatives simply due to a longer time since survey completion? 

Thank you, we have updated the median and IQR to present these after excluding those who have recovered (in the abstract and the results section). The mean and SD are unchanged. We have also updated the n in Table 4 where we have presented mean duration of illness.

Yes, the duration of illness was calculated as the time from symptom onset to the date of survey completion for those still symptomatic. 

The longer duration among negatives/not tested is likely to reflect earlier symptom onset when access to testing (in the UK) was limited.

7. Table 1 and 2 footnote: revise to “Comparisons between those with and without lab-confirmed COVID-19 used...”

Thank you, we have revised the footnote for Tables 1-6 as suggested.

8. Results – previous health: revise last sentence to “There were no significant differences in these proportions between those with and without lab-confirmed infection”.

Thank you, we have revised as suggested.

9. Results, course of illness: revise line 287-288 of tracked version to “…which were higher in those with lab confirmation than without, whereas abdominal pain, nausea, chest pain, chest tightness, chills, hoarse voice, sore throat, sneezing and pins and needles were lower in those with lab-confirmation than without.”

Thank you, we have revised as suggested.

10. Table 4: ‘Comes and goes’ and ‘relapsing’ are listed separately in the table but reported/defined together in the text (line 314). How are these different?

Thank you, we have combined these into one category.

 ‘Comes and goes’ was one of the options included in our survey but we also gave participants an ‘other’ option which they could choose and describe the pattern of their illness if they didn’t feel the available options captured the pattern they experienced. The responses to the other option were then categorised by the research and coded as a separately relapsing category.

11. Results – lab confirmation of infection: Move sentence (starting with “1172 participants (46%) reported having an antibody test…”) on line 381 of tracked version to the beginning of that paragraph on antibody testing. Also add a period between ‘test’ and ’26.3%’ on line 379.

Thank you, we have revised as suggested.

12. Table S4 – many of the categories are overlapping – please revise.

Thank you, we have updated this to clarify the categories. 

13. Results – lab confirmation of infection: the last paragraph seems redundant with the prior sections since differences by lab confirmation are now presented throughout – suggest deleting it. 

Thank you, we have moved two of the sentences that were not previously covered in the text and deleted the rest of the paragraph as suggested.

14. Results, clustering: second paragraph describing results of univariate analysis – for some the % are presented in the text (e.g. for loss of income)- suggest removing to be consistent for all results mentioned.

Thank you, we have revised as suggested.

15. Results, clustering: last paragraph, correct reference to “OR” for ‘having a confirmed positive test’.

Thank you, we have corrected this as suggested.

---

## [Editor Report · Decision Letter 2]

9 Feb 2022

Characteristics and impact of Long Covid: findings from an online survey

PONE-D-21-14682R2

Dear Dr. Ziauddeen,

We’re pleased to inform you that your manuscript has been judged scientifically suitable for publication and will be formally accepted for publication once it meets all outstanding technical requirements.

Kind regards,

Catherine G. Sutcliffe

Academic Editor

PLOS ONE
---

## [Editor Report · Acceptance letter]

16 Feb 2022

PONE-D-21-14682R2 

Characteristics and impact of Long Covid: findings from an online survey 

Dear Dr. Ziauddeen:

I'm pleased to inform you that your manuscript has been deemed suitable for publication in PLOS ONE. Congratulations! Your manuscript is now with our production department. 

Kind regards, 

on behalf of

Dr. Catherine G. Sutcliffe 

Academic Editor

PLOS ONE